# Single-cell morphometrics reveals T-box gene-dependent patterns of epithelial tension in the Second Heart field

Clara Guijarro[1,2,3,4], Solène Song[2,3,4], Benoit Aigouy[1], Raphaël Clément [1], Paul Villoutreix [2,3,5] ✉ & Robert G. Kelly [1,5] ✉

The vertebrate heart tube extends by progressive addition of epithelial second heart field (SHF) progenitor cells from the dorsal pericardial wall. The interplay between epithelial mechanics and genetic mechanisms during SHF deployment is unknown. Here, we present a quantitative single-cell morphometric analysis of SHF cells during heart tube extension, including force inference analysis of epithelial stress. Joint spatial Principal Component Analysis reveals that cell orientation and stress direction are the main parameters defining apical cell morphology and distinguishes cells adjacent to the arterial and venous poles. Cell shape and mechanical forces display a dynamic relationship during heart tube formation. Moreover, while the T-box transcription factor Tbx1 is necessary for cell orientation towards the arterial pole, activation of Tbx5 in the posterior SHF correlates with the establishment of epithelial stress and SHF deletion of Tbx5 relaxes the progenitor epithelium. Integrating findings from cell-scale feature patterning and mechanical stress provides new insights into cardiac morphogenesis.

During embryonic development, cells adopt diverse strategies to sculpt epithelial tissues into organs with molecularly and mechanically distinct regions and characteristic 3D shapes. Organogenesis of the vertebrate heart, originating in epithelial cells in anterior splanchnic mesoderm of the early embryo, is a good example. The cardiac primordium is formed by First Heart Field (FHF) progenitor cells generated by an early wave of mesodermal cell commitment to cardiac fate[1–3]. The heart tube then grows by the progressive addition of Second Heart Field (SHF) progenitor cells from contiguous splanchnic mesoderm in the dorsal pericardial wall (DPW). While SHF cells initially contribute along the entire length of the cardiac primordium, progenitor cell addition becomes restricted to the arterial and venous poles of the heart after breakdown of the dorsal mesocardium and formation of the ventral heart tube[4]. SHF cells ultimately give rise to myocardium of the right ventricle and outflow tract at the arterial pole,

and atrial myocardium, including atrial septal structures, at the venous pole. As SHF progenitors incorporate, the heart tube elongates, buckles and loops to the right[5,6].

Heart tube extension by progenitor cell deployment creates the template for subsequent cardiac septation events that generate the definitive four-chambered heart. Thus perturbation of SHF addition leads to a spectrum of congenital heart defects including outflow tract and atrial septal defects, observed in human genetic disorders such as 22q11.2 deletion (or DiGeorge) syndrome[7], and Holt-Oram syndrome[8,9]. The T-box transcription factor encoding genes *TBX1* and *TBX5* are implicated in these syndromes, respectively, and have been shown in mouse models to play major roles in regulating progenitor cell addition to the poles of the heart and early heart morphogenesis[10–13]. Patterning of the SHF epithelium leads to the establishment of distinct TBX1 expressing arterial pole progenitor cells

[1]Aix-Marseille Université, CNRS UMR 7288, IBDM, Turing Centre for Living Systems, Marseille, France. [2]Aix-Marseille Université, LIS, UMR 7020, Turing Centre for Living Systems, Marseille, France. [3]Aix-Marseille Université, MMG, Inserm U1251, Turing Centre for Living Systems, Marseille, France. [4]These authors contributed equally: Clara Guijarro, Solène Song. [5]These authors jointly supervised this work: Paul Villoutreix, Robert G. Kelly. ✉ e-mail: Paul.Villoutreix@univ-amu.fr; Robert.Kelly@univ-amu.fr

in the anterior SHF and TBX5 expressing venous pole progenitors in the medial posterior SHF. TBX1 coordinates cell addition to the arterial and venous poles[14,15]. It also regulates proliferation, differentiation and epithelial polarity in the SHF[16,17] while TBX5 is required for left ventricular and venous pole development, atrial septation and cell cycle progression in the posterior SHF[9,11,12]. Despite this clinical significance the cellular mechanisms by which contiguous epithelial progenitor cells are incorporated into the cardiac primordium are poorly understood.

Apicobasal polarity studies revealed that the SHF is an atypical polarized epithelium. Cell contacts and epithelial properties are restricted to the apical region of SHF cells while the basolateral membrane forms dynamic filopodia-like protrusions more commonly observed in mesenchymal cells[17]. Evidence for epithelial tension in the DPW has suggested that SHF cells are subject to mechanical forces that contribute to the process of heart tube extension[18]. In support of such a model, oriented cell growth and cell division together with ppMLC and YAP/TAZ accumulation have been observed in the posterior DPW and wounding experiments have been performed on the epithelium showing oriented deformation consistent with epithelial tension[19]. Further evidence for the importance of epithelial properties of the SHF has emerged from studies of the Planar Cell Polarity (PCP) pathway. Perturbation of *Wnt5a*, encoding a non-canonical Wnt ligand regulating PCP and a transcriptional target of TBX1, impacts SHF addition to the arterial and venous poles, leading to both outflow tract and atrial septation defects[20,21]. In the absence of the PCP gene *Vangl2*, SHF-derived cells are abnormally polarized and disorganized, which leads to a shortening of the OFT and loss of SHF cell progenitor status[22]. In addition, N-cadherin and Fibronectin expression in the SHF are required for epithelial cell architecture and normal heart tube elongation[23,24]. While these studies highlight the importance of epithelial properties of the SHF for progenitor cell deployment, the mechanisms and cues involved in the establishment and regulation of cardiac progenitor cell addition to alternate poles of the heart remain unknown.

Studies more focused on the physical aspects of epithelial morphogenesis have shown that animal cells can actively change their shape through their contractile activity as well as being passively shaped by forces transmitted across the tissue within which they reside[25]. Indeed, contractile forces can propagate across long distances in the plane of the epithelium. Such forces can be counter-balanced by outward pressure generated by tissue growth. Therefore, morphometric analysis and evaluation of mechanical forces are, in general, essential to make sense of morphogenetic movements and organogenesis[26–30]. Assessing the mechanical state of cells in the SHF is therefore of major interest for understanding how the embryonic heart acquires its shape.

The development of image analysis approaches to segment and quantify apical epithelial parameters[31,32] as well as mechanical features[33] at the level of individual cells, has led to the generation of large datasets. The high degree of complexity of these datasets calls for the use of unsupervised data analysis approaches[34] to explore individual cell features. In addition, the spatial aspect of these types of datasets calls for the development of specific methods that distinguish spatial regions of cells with related features. In a similar way, approaches have been applied to the analysis of large single-cell RNA sequencing datasets to generate tissue gene expression cartography[35]. Previous studies of epithelial features of the SHF epithelium have focused on manually selected regions of the anterior and posterior DPW[19,20]. However, the choice of selected regions might not be representative of the entire tissue level dynamics. A quantitative analysis of morphological and mechanical features at the single-cell level across the entire epithelium coupled with unsupervised approaches such as dimensionality reduction and clustering should reveal dynamic features associated with cell deployment in the progenitor cell epithelium.

The purpose of our study is to create a quantitative map of the morphological and mechanical properties of cardiac progenitors in the DPW and investigate how T-box genes impact on these features to orchestrate the addition of progenitor cells to the cardiac poles. To achieve this aim, we quantified cell morphometrics and inferred forces at single-cell level in the entire SHF epithelium. Once these single-cell features were obtained, we adapted a spatial PCA approach[36] to characterize the major axes of variation and perform unsupervised clustering analysis of cell features. Our results reveal that cell orientation and the principal direction of cell stress are the main features explaining both the variation between cells and the spatial coherence of apical cell morphology in the DPW at embryonic day (E) 9.5. Single-cell morphometric analysis prior to dorsal mesocardium breakdown confirms that patterned epithelial tension across the DPW arises as SHF addition become restricted to the cardiac poles. We show that the T-box transcription factors TBX1 and TBX5 differentially impact on tissue stress patterns. Arterial pole progenitor cell behavior correlates with *Tbx1* expression which is required to orient cells around the outflow tract and create a region with low isotropic stress. In contrast, activation of *Tbx5* in the posterior medial region of the DPW between E8.5 and E9.5 correlates with the acquisition of epithelial stress in the DPW. Mutant analysis reveals that Tbx5 activation is required to orient cell shape and cell stress parallel to the antero-posterior axis in venous pole progenitor cells. Together these results reveal that single-cell morphometrics can provide insights into how patterned epithelial stress is established in the SHF during early cardiac organogenesis, revealing the interplay between genetics and mechanics during organogenesis.

## Results
### Single-cell morphometric analysis of the second heart field

The DPW is a heterogeneous epithelial tissue with an organized cellular architecture. Morphogenesis results from dynamic interactions between patterning signals, cell regulatory networks, and tissue geometry[37]. Previous studies have suggested the importance of epithelial tension in the SHF for the growth of the heart[18–20]. To further investigate the importance of cell morphological features and cell mechanics in SHF deployment, we have developed a new spatial single-cell morphometric analysis pipeline (Fig. 1a). To perform single-cell morphometrics, we built a dataset of morphological and mechanical features calculated from Tissue Analyzer[32], an ImageJ package to segment, track and quantify 2D epithelial tissues, Deproj, a MATLAB package to yield accurate morphological measurements on 2D projections[31] and Bayesian Force Inference, a MATLAB package to quantify and infer the distribution of forces in a developing tissue from their shape and connectivity[33,38] (Fig. 1a, see "Methods" section). Frontal whole-mount views of microdissected DPWs of 6 E9.5 wild-type embryos staged between 19-24 somites were analyzed with a total of 8618 cells. Apical cell contours were marked by phalloidin staining. 9 features related to cell morphology and cell mechanics were calculated for each cell of the epithelia. Z-stacks were imaged covering the entire depth of the epithelia (40-50 μm depth) from the apical to the basal side of the cells, allowing us to generate a 3D acquisition of the cells in the epithelium, used to obtain the 2D projection.

The 9 features comprised 6 morphological features and 3 mechanical features (Fig. 1b, c). The morphological features evaluated were: (1) orientation of the main cell elongation axis, (2) apical cell area, (3) apical cell perimeter, (4) cell eccentricity (apical circularity index), a measure of the elongation of cell shape. In addition, we scored the number of cell neighbors (number of cell apical sides) (5) as a characteristic of cell shape within epithelia[39]. We also evaluated nucleus-to-Golgi polarity (6), which is the angle formed between the vector from the center of the nucleus to the center of the Golgi, relative to the embryonic left-right axis. To measure epithelial stress patterns, we used a force inference MATLAB plug-in[33]. This image analysis-based approach infers forces, including stress patterns, in epithelial tissues

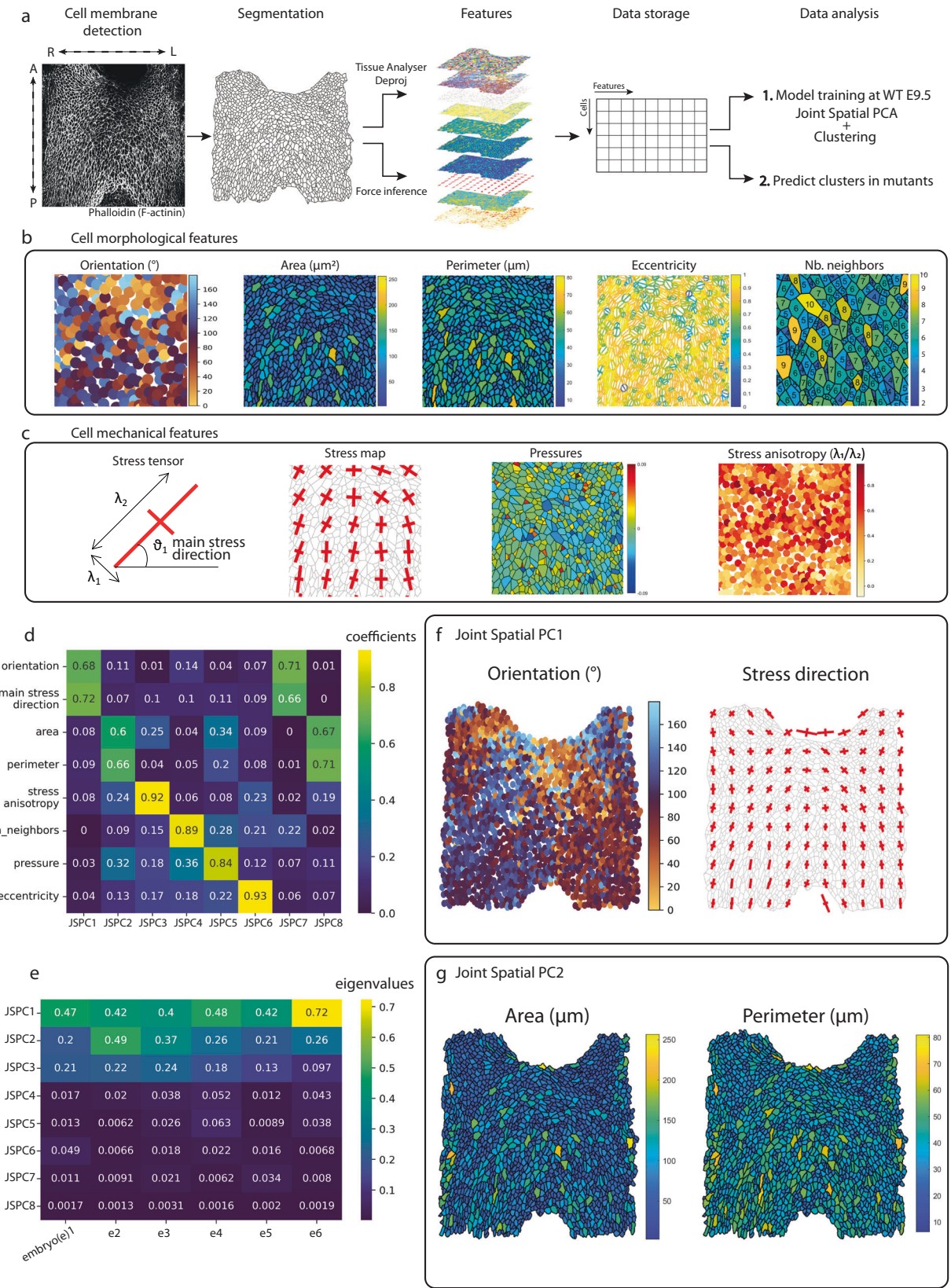

**Fig. 1 | Spatial single-cell morphometrics highlights cell orientation, stress direction and size as the main parameters defining apical cell morphology in the dorsal pericardial wall. a** Workflow for the single-cell morphometric analysis of cells in the dorsal pericardial wall. **b**, **c** Description of the cell features used for the single-cell morphometrics. **d** Heatmap showing the coefficients for each feature that compose each joint spatial principal component axis (jsPCA). **e** Heatmap showing the eigenvalues for each principal component. **f** Visualization of the main joint spatial PCA1 parameters: cell orientation and stress tensor in real space. **g** Visualization of the main jsPCA2 parameters: cell area and cell perimeter in real space. A anterior, P posterior, R right, L left, λ stress amplitude, ϑ stress main orientation, e embryo, JSPC Joint spatial principal component. $N = 6$ embryos, $n = 8618$ cells. Source data are provided as a Source Data file.

based on the topology and geometry of cell contacts[33,38]. The three mechanical features we analyzed were cell pressure, the principal direction of the cell stress tensor ($\vartheta_1$) and stress anisotropy, which is the ratio of the amplitudes of the stress tensor in the principal and secondary directions ($\lambda_1/\lambda_2$).

Nucleus-to-Golgi polarity was included as a potential read-out of planar polarity in DPW cells. In front-rear cell polarity models, such as directed cell migration, centrosomal reorientation towards the leading edge is known to align the Golgi apparatus towards the leading edge[40,41]. Potentially, in the DPW, the Golgi may be oriented towards the arterial or the venous pole. The nucleus-to-Golgi morphological feature was also addressed in 3D reconstructions (Fig. S1). The results show that 90% of cells in the DPW have the Golgi positioned apically to the nucleus. However, comparison of the position of the Golgi in the anterior-posterior or left-right axes revealed no preferred localization for the Golgi relative to the nucleus in cells from different regions of the DPW. The Golgi is located in the anterior part of the cell in 50% of the cells and in 50% of the cells Golgi is located in the right part of the cell (Fig. S1). This suggests that apicobasal polarity rather than planar polarity determines Golgi position in DPW cells.

We performed principal component analysis (PCA) on these measurements for cells in the E9.5 DPW. PC1 (Principal component 1) reflects cell size (cell area and perimeter) and cell pressure; PC2 is a combination of cell orientation and cell stress direction; PC3 is composed mainly of stress anisotropy and cell eccentricity while PC4 is composed mainly of number of neighbors (Fig. S2). PC1 to PC4 are the major contributors to variation: PC1 explains 29% of the heterogeneity of the cells in the DPW according to their morphological and mechanical characteristics, PC2 explains 19% of the variability of the cells, PC3 13% and PC4 11% (Fig. S2). While PCA is an adapted tool to characterize the intrinsic variation of single-cell features, we investigated whether it also accounted for the spatial distribution of cells within the tissue sample. To assess this, we computed the spatial-autocorrelation of each principal component, also known as Moran's Index (MI). We found that PC1 had a lower MI than PC2, and PC2 had the highest MI across all PCs (Fig. S2). Analysis of correlations between these parameters shows that cell area and perimeter have a correlation coefficient of 0.9 and stress direction and cell orientation have a correlation of 0.7, while the remaining features have lower correlation coefficients (Fig. S3). While PCA focuses mainly on feature covariation, our analysis suggests that it is not sufficient to capture the spatial variation within the data.

### Development of joint spatial PCA of sc-morphometric data

We investigated if we could identify meaningful spatial patterns for the single-cell morphological and mechanical features within the tissue. For this, we turned to spatial PCA[36], a dimensionality reduction method that identifies the main directions of variation within the data, while considering the spatial distribution. To obtain this double information, we built on a method considering the product matrix between the covariance of the data and Moran's Index. The eigenvectors of this product matrix and their associated eigenvalues reveal the directions that maximize both the covariance and the spatial autocorrelation of linear combinations of variables. This approach generalizes the classical PCA, which aims to identify the major direction within the covariance matrix[36]. As shown by Jombart et al.[36], the product matrix between the covariance matrix and Moran's Index can be rewritten in a simple form as a function of the feature matrix and the adjacency matrix, a matrix encoding how close cells are within the tissue. The gain in information by considering jointly the variance in the data and the spatial information comes, however, at the expense of the ability to combine separate tissue samples. Specifically, the product matrix is dependent on the adjacency matrix which is sample-specific and cannot be easily pooled between samples. In order to consider jointly samples in the same conditions, we performed a joint diagonalization[42] of the product matrix (see Methods for details). This approach, which

we name joint spatial PCA (jsPCA), yields the joint spatial Principal Components of features within a collection of tissues.

### Patterned cell orientation and stress direction in the SHF

We applied the joint spatial PCA pipeline to the single-cell morphometric dataset of cells in the DPW of E9.5 WT embryos. The first joint spatial Principal Component (jsPC1) for each of the sample was composed of cell orientation and stress direction, while the second joint spatial Principal Component (jsPC2) was composed of cell area and cell perimeter (Fig. 1d). These first two jsPC capture most of the variation for all embryos of the cohort (Fig. 1e). Since the aim of jsPCA is to obtain orthogonal directions of variations, we expect the patterns associated with jsPC1 and jsPC2 to be quite different, as represented in Fig. 1f,g. The jsPCA approach was then used to perform dimensionality reduction by projecting the multivariate distribution of cells of each sample on jsPC1. We used the projected data to perform an unsupervised clustering using a Gaussian mixture approach. We found that using 3 clusters provided a relevant way of representing the data which is consistent with the minimization of the Akaike Information Criterion (Fig. S4). In addition, we were able to use the cluster characteristics learned on E9.5 WT embryos to define comparable clusters in other conditions.

Mapping back cells of the three clusters onto the E9.5 WT embryo revealed that cells in clusters 1 and 2 are distributed in a bilaterally symmetrical manner on the right and left sides of the DPW, alternating around the inferior outflow tract in the anterior DPW, while cells labeled as cluster 3 are predominantly located in the posterior part of the DPW (Figs. 2a,b, S4, and S5). As cell stress direction, orientation, area, and perimeter are the main features according to the joint spatial principal component analysis (jsPCA), we calculated the mean values for the 4 parameters for each of the 3 clusters. These mean values serve as feature signatures defining each cluster. Thus, the morphological signatures of the cells labeled as cluster 1 are: cell stress direction: $45° \pm 20$ and cell orientation: $45.78°\pm20$, cell area: $54.4\,\mu m^2 \pm 32$ and cell perimeter: $33.06\,\mu m \pm 9$. The morphological signatures for cells labeled as cluster 2 are: cell stress direction: $-47.1° \pm 20$ and cell main orientation: $-47.7° \pm 20$, cell area: $47.7\,\mu m^2 \pm 27$ and cell perimeter: $30.7\,\mu m \pm 9$. The morphological signatures for cells labeled as cluster 3 are: cell stress direction: $1.46° \pm 28$ and cell stress main orientation: $-0.57° \pm 29$, cell area: $51.66\,\mu m^2 \pm 30$ and cell perimeter: $32.34\,\mu m \pm 10$ (Fig. 2c–f). Mean values for the remaining features are common to clusters 1–3 (Fig. 2g–j). Our single-cell morphometrics analysis thus permits visualization of patterned epithelial features indicative of patterned tension across the DPW at the stage of heart tube elongation. Given the bilateral symmetry of clusters 1 and 2 we also classified cells into two clusters computed with cell orientation and stress direction symmetrized with respect to the anterior-posterior axis (Figs. 2k and S4). Although introducing a priori knowledge about the symmetry of the system, and hence some level of supervision, this two cluster analysis highlights the distinct stress patterns in the anterior and posterior DPW and the radial symmetry of anterior DPW cells as they converge on the arterial pole.

### Patterned tension in the SHF emerges as the heart tube forms

The connection between the DPW and the heart tube varies over time: at E8.5, the DPW epithelium is connected to the early heart by a mesentery, the dorsal mesocardium, along the antero-posterior axis (Fig. 3a). After the dorsal mesocardium breaks down behind the heart tube, the bilateral right and left sides of the DPW epithelium fuse. As a result, the heart tube remains connected to the SHF only at the arterial and venous poles (Fig. 3b). Single-cell morphometrics was performed on whole-mount DPWs microdissected from E8.5 wild-type embryos with 6-8 somites, before breakdown of the dorsal mesocardium (Fig. 3c). The clusters learnt in the E9.5 wild-type embryos were used to classify E8.5 DPW cells.

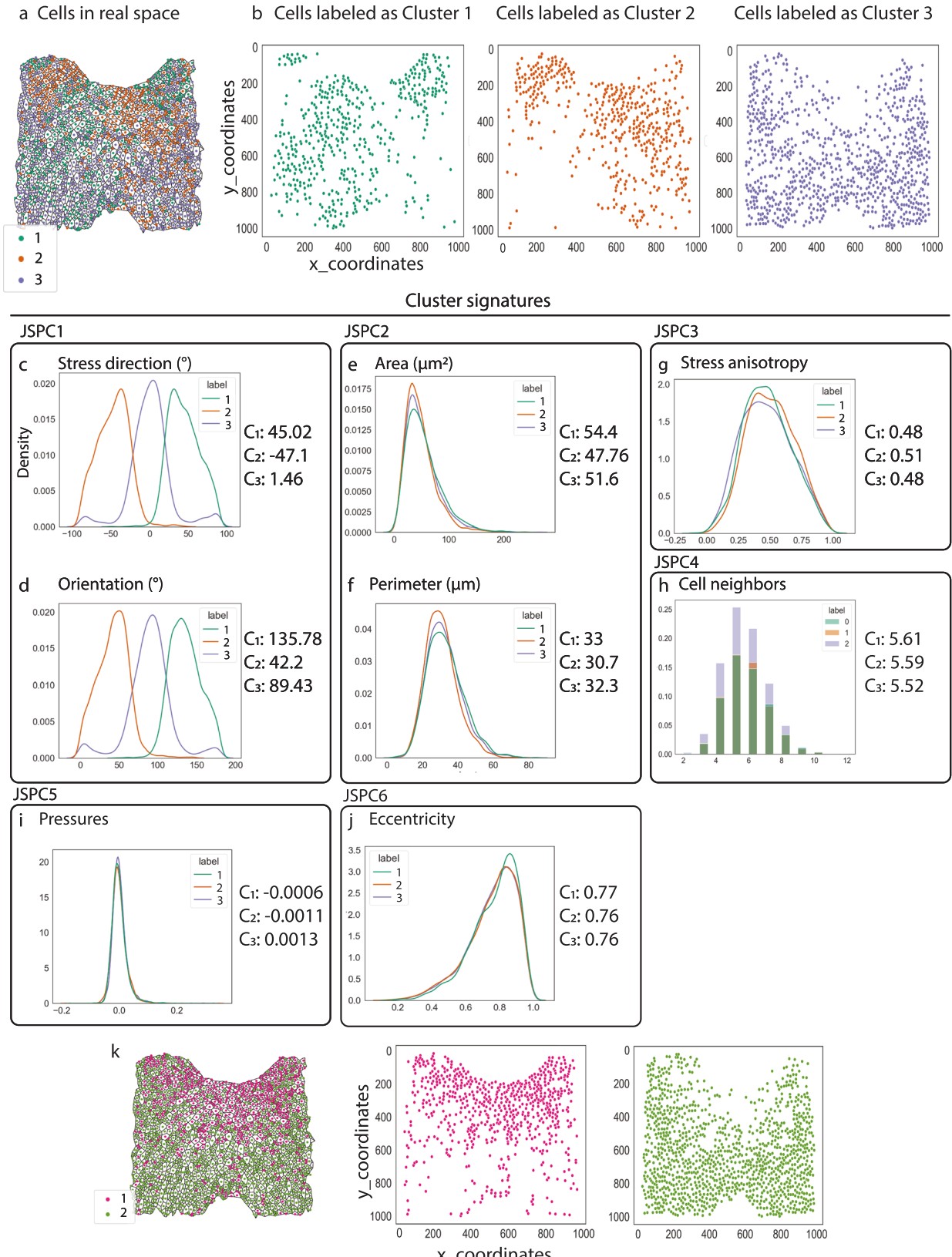

**Fig. 2 | Spatial single-cell morphometrics define regions in the DPW based on morphological and mechanical cell properties. a**. E9.5 wild-type cells plotted back to their physical space and color coded with the cluster number; cluster 1 cells are in green, cluster 2 cells are in orange and cluster 3 cells are in purple. **b** Scatter plots of the cells by their cluster label in the real space. **c–j** Kernel density estimate plots of the different features defining each cluster label. The measurements for cells belonging to cluster 1 are in green, for cluster 2 are in orange and for cluster 3 in purple. **k** E9.5 wild-type cells plotted back to their physical space and color coded after classification into two clusters following symmetrization of cell orientation and stress direction with respect to the anterior-posterior axis. Cluster 1 cells are in pink and cluster 2 cells are in green. $N = 6$ embryos, $n = 8618$ cells.

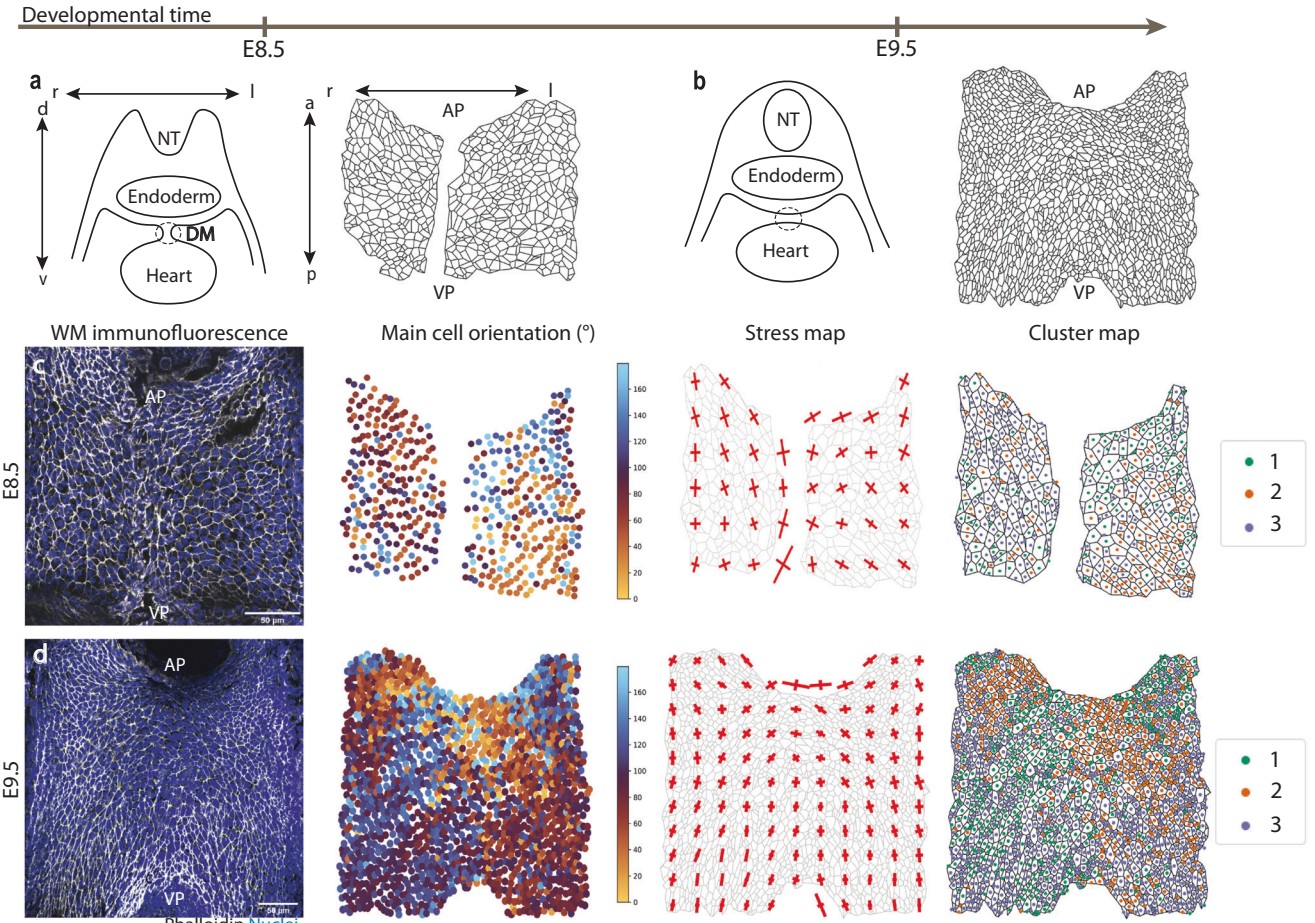

**Fig. 3 | Spatial single-cell morphometrics reveals that patterned tension in the second heart field emerges during heart tube formation. a, b**. Scheme of the architecture of the SHF at embryonic days 8.5 (**a**) and 9.5 (**b**) showing breakdown of the dorsal mesocardium (DM) between these timepoints. **c** Frontal view of E8.5 wild-type whole mount dorsal pericardial wall, main cell orientation map, stress map and cluster map. **d** Frontal view of E9.5 wild-type whole-mount dorsal pericardial wall, main cell orientation map, stress map and cluster map. Scale bars: 100 μm. NT neural tube, AP arterial pole, VP venous pole, r right, l left, d' dorsal, v ventral, a anterior, p posterior. Wild-type E8.5: N = 6 embryos, n = 2994 cells; wild-type E9.5: N = 6 embryos, n = 8618 cells.

Quantitative evaluation of apical cell properties at E8.5 revealed a striking difference in the orientation of cells in the posterior DPW with respect to that at E9.5, before and after dorsal mesocardium breakdown. At E8.5 DPW cells are oriented parallel to the embryonic left-right axis and under isotropic stress (Figs. 3c and S6). In contrast, at E9.5, cells in the posterior DPW, both medially and laterally, are oriented at 90° ± 10, and display stress anisotropy with the main stress tensor oriented along the anterior-posterior embryonic axis (Fig. 3d). This reveals the emergence of an E9.5-specific cluster enriched in the posterior DPW between these timepoints, consistent with analysis of cell elongation in selected subregions of the DPW. While in late E9.5 embryos with >20 somites cluster 3 is restricted to the posterior DPW, in embryos with <20 somites an enrichment of cells belonging to cluster 3 was observed in the midline of the DPW, the location of dorsal mesocardial breakdown, expanding towards the anterior DPW (Fig. S5e, f). Classification of E8.5 cells into two clusters, computed with symmetrized cell orientation and stress direction, also revealed that distinct anterior and posterior clusters emerge between E8.5 and E9.5 (Fig. S6). These results reveal the dynamic nature of cell shape changes and patterns of epithelial tension in the DPW during these stages of heart tube morphogenesis. In particular, they highlight the emergence of epithelial features associated with elevated tension in the posterior DPW after breakdown of the dorsal mesocardium.

## Comparison of cluster and T-box gene distribution in the SHF

Previous studies have shown, in serial sagittal sections, the emergence of a transcriptional boundary delimiting TBX1 and TBX5 positive domains in the DPW after the dorsal mesocardium breaks down. These domains arise by de novo activation of *Tbx5* expression in the medial posterior region of the DPW at E8.5 followed by TBX1-dependent downregulation of the anterior SHF program in TBX5 positive cells by E9.5[14]. We validated *Tbx1* and *Tbx5* expression domains using RNA-scope fluorescent in situ hybridization on frontal whole-mount views of the DPW and quantification of mRNA intensity (Figs. 4a–e and S7). At early stages (6-8 somites), the DPW is entirely *Tbx1*-positive (Fig. 4a, b). As the dorsal mesocardium breaks down behind the central part of the heart tube, *Tbx5* expression is activated in medial posterior SHF cells close to the venous pole (Fig. 4c). *Tbx1* and *Tbx5* are transiently co-expressed (Fig. 4d) prior to the establishment of a boundary as *Tbx1* is downregulated in *Tbx5* expressing cells. At E9.5 distinct *Tbx1+* and *Tbx5+* domains of the DPW are well defined (Fig. 4e). *Tbx1+ Tbx5-* cells are observed in the anterior DPW and lateral regions of the posterior DPW. Comparison of *Tbx1* and *Tbx5* expression at E9.5 with the sc-morphometric cluster distribution reveals that *Tbx1* expression correlates largely with the distribution of clusters 1 and 2. In contrast, *Tbx5* expressing cells in the posterior DPW at E9.5 fall entirely within cluster 3, that also encompasses *Tbx1+Tbx5-* cells in the lateral regions of the posterior DPW (Fig. 4f).

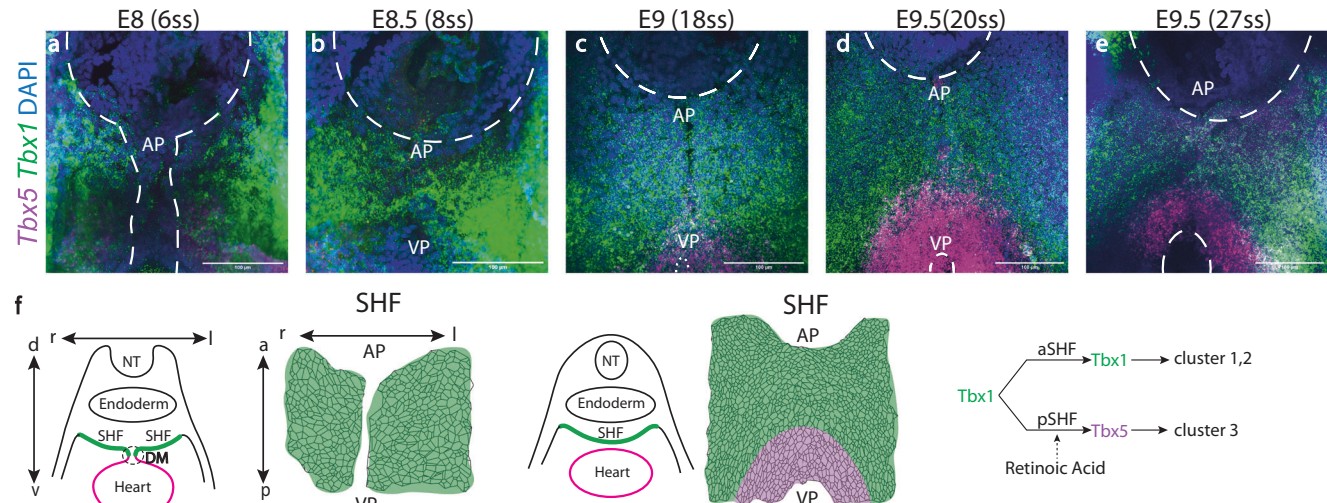

**Fig. 4 | Clusters defined by spatial single-cell morphometrics correlate with the distribution of T-box transcription factors in the dorsal pericardial wall.** **a–e** Frontal whole-mount views of the DPW after RNAscope fluorescent in situ hybridization showing *Tbx1* (green) and *Tbx5* (magenta) mRNA distribution between E8 (6 somites) and E9.5 (25 somites), before and after dorsal mesocardial breakdown. Nuclei are stained using Hoescht (blue) (**f**) Summary of the expression profiles of *Tbx1* and *Tbx5* based on the schemas in Fig. 3a, b, showing the links between *Tbx1* and *Tbx5* expression domains and clusters 1–3, and the input of retinoic acid signaling[14]. AP arterial pole, VP venous pole, aSHF anterior second heart field, pSHF posterior second heart field. E8-E8.5: *N* = 5 embryos, E9: *N* = 3 embryos, E9.5: *N* = 6 embryos. Scale bars: 100 μm.

## TBX1 and TBX5 pattern apical cell architecture in the SHF

*Tbx1* and *Tbx5* are essential for normal development of the arterial and venous poles of the developing heart, respectively. Moreover, haploinsufficiency for *TBX1* or *TBX5* contributes to congenital defects affecting the arterial and venous poles of the heart in 22q11.2 deletion and Holt-Oram syndromes[43,44]. As the emergence of distinct *Tbx1* and *Tbx5* positive territories in the DPW coincides with the breakdown of the dorsal mesocardium, and the expression profile of these genes correlates with the distribution of cells from clusters 1 and 2 (*Tbx1*) and cluster 3 (*Tbx5*), we used single-cell morphometrics to investigate the impact of TBX1 and TBX5 loss of function on patterns of apical cell morphology in the DPW. To this end we performed single-cell morphometric analysis of the DPW of E9.5 *Tbx1* homozygous null (*Tbx1-/-*) embryos[7,45,46] in which SHF cell deployment to the arterial pole is defective, and conditional *Tbx5* mutant (*Mef2cAHFCre;Tbx5fl/fl*) embryos[11,47,48] in which *Tbx5* expression is deleted in the SHF without affecting *Tbx5* in the first heart field and early heart tube. SHF deletion of *Tbx5* results in defective posterior SHF contributions to the venous pole, and later failure of atrial and ventricular septal morphogenesis at the heart field interface[11,47]. Single-cell morphometrics was performed on whole-mount DPWs microdissected from *Tbx1* mutant and *Tbx5* conditional mutant embryos at E9.5. As for the E8.5 dataset, cells from mutant embryos were classified following the clusters learnt from the E9.5 wild-type dataset (Figs. 5 and S8–10).

In *Tbx1-/-* embryos, the DPW is hypoplastic and less organized. Single-cell morphometric analysis reveals a loss of the two main bilateral populations, corresponding to clusters 1 and 2, in the aDPW of *Tbx1* mutant embryos (Figs. 5a', b' and S9). Cells of cluster 3 are enriched in the posterior DPW, as in E9.5 wildtype embryos, characterized by cells oriented at 90°, stressed parallel to the embryonic anterior-posterior axis and with mean cell area and perimeter of 50μm² and 35 μm respectively (Fig. 5a',b'). In contrast, in *Mef2cAHFCre;Tbx5fl/fl* embryos, cluster 3, which is regionalized at the venous pole in wild type, is no longer enriched in the posterior DPW and distributed throughout the epithelium (Figs. 5c', S10). The 2 remaining clusters (clusters 1 and 2) correspond to the left and right sides of the DPW of *Mef2cAHFCre;Tbx5fl/fl* embryos. Quantification of the distribution of cells along the anterior-posterior axis of the DPW revealed significant posterior enrichment of cluster 3 cells in wildtype E9.5 and *Tbx1-/-*

embryos, but not in wildtype E8.5 or *Mef2cAHFCre;Tbx5fl/fl* embryos (Fig. 5d). Single-cell morphometric analysis of cells in the DPW of heterozygous *Tbx5* mutant embryos (*Mef2cAHFCre;Tbx5fl/+*) revealed similar results to the wild-type situation (Fig. S11). Classification of *Tbx1-/-* and *Mef2cAHFCre;Tbx5fl/fl* cells into two clusters, computed with symmetrized cell orientation and stress direction, revealed that while two regionalized clusters are maintained in *Tbx1-/-* embryos, emergence of a distinct posterior DPW cluster fails in *Mef2cAHFCre;Tbx5fl/fl* embryos (Fig. S9 and 10). Together these results demonstrate that *Tbx1* and *Tbx5* differentially impact on patterns of apical cell morphology in the DPW and reveal that activation of *Tbx5* in the SHF is necessary for the regionalization of cluster 3 cells in the posterior DPW at E9.5.

## Tbx5 regulates tension and orientation in the posterior SHF

In order to further understand how the absence of TBX1 and TBX5 differentially impact apical cell morphology in the DPW we examined the individual parameters constituting the single-cell morphometric datasets, compared to E8.5 and E9.5 wildtype embryos (Fig. 6a, b). In *Tbx1* mutant embryos a larger TBX5+ domain is observed in the DPW (Fig. 6c), characterized by cells oriented at 90° as in wildtype embryos, as a result of the loss of the two bilateral domains of clusters 1 and 2 in the aDPW. In conditional *Tbx5* mutant embryos, the TBX5+ domain in the DPW is absent and cells in both the anterior and posterior DPW are oriented in a bilaterally symmetric manner towards the arterial pole (Fig. 6d). Cells oriented parallel to the antero-posterior axis are not regionalized in the posterior DPW of SHF *Tbx5* mutant embryos but are instead observed to be scattered throughout the DPW, both anteriorly and posteriorly (Fig. 6d). Quantification of cells oriented horizontally (0° ± 10, along the embryonic left-right axis) and cells oriented vertically (90°, along the embryonic anterior-posterior axis), reveals an enrichment of horizontal cells in the anterior DPW and an enrichment of vertical cells in the posterior DPW in the presence of TBX5, but not in E8.5 embryos prior to activation of *Tbx5* in the DPW or in *Tbx5* SHF mutant embryos at E9.5 (Fig. 6e).

A biomechanical feedback loop has been suggested to contribute to elongation of the heart tube through epithelial tension in cardiac progenitor cells in the posterior DPW[18]. As the joint spatial PCA algorithm highlights stress direction as one of the main parameters that

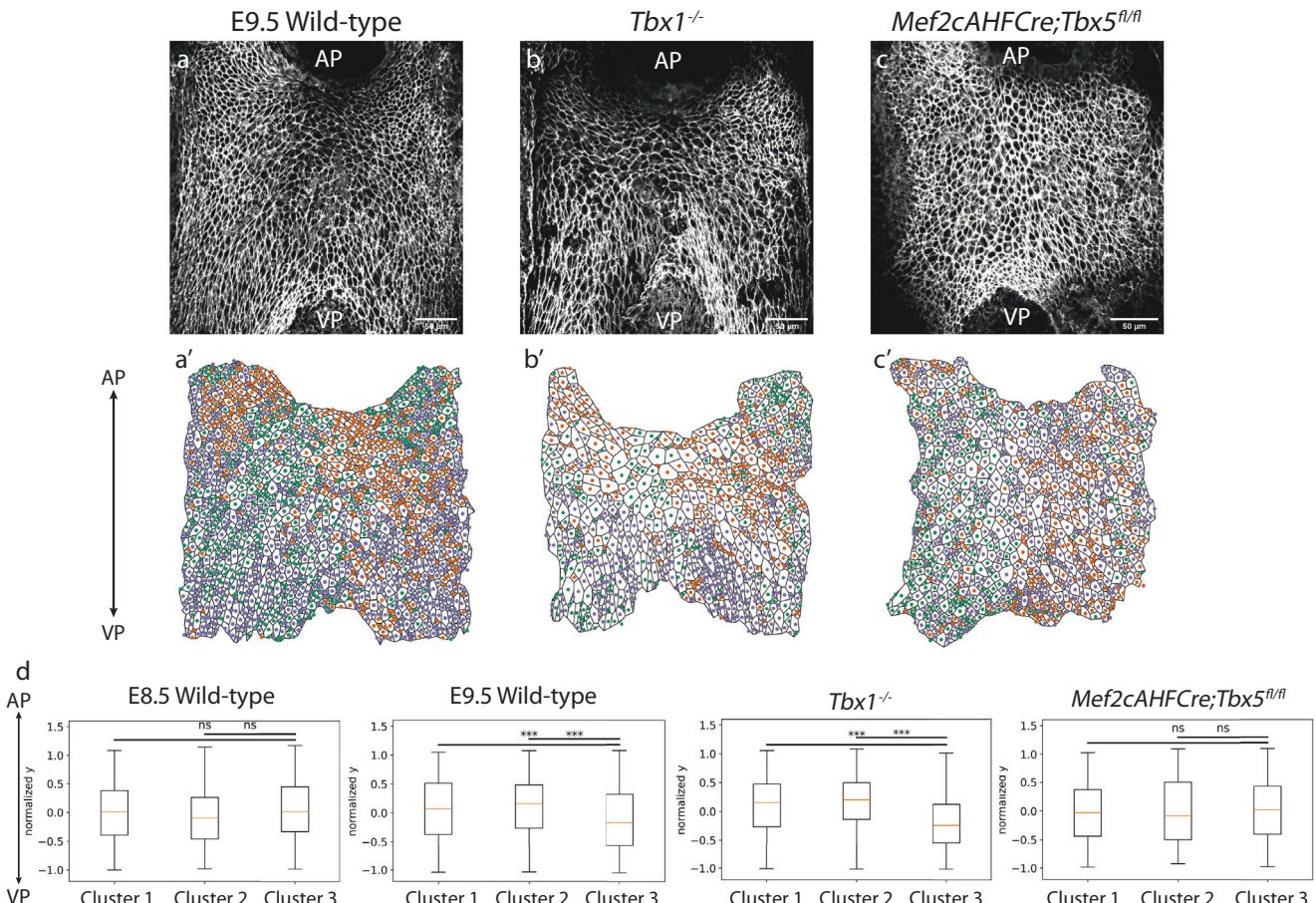

**Fig. 5 | T-box transcription factors TBX1 and TBX5 pattern apical cell architecture in the dorsal pericardial wall. a–c** Ventral whole-mount views of E9.5 wild-type, *Tbx1*[-/-], and *Tbx5* conditional mutant DPWs stained with phalloidin. **a′–c′**. Cell segmentation of E9.5 wild-type, *Tbx1* mutant, and *Tbx5* conditional mutant and cells labeled according to their cluster number (cluster 1 green, cluster 2 orange, cluster 3 purple). **d** Boxplot quantifications of the spatial coordinates of cells belonging to each of the clusters along the antero-posterior axis (with y coordinates normalized from -1 (VP) to 1 (AP)) for E8.5 ($p = 0.96$, $p = 1$) and E9.5 wild-type (***$p = 5.89e-35$,

***$p = 5.91e-64$), *Tbx1*[-/-] (***$p = 5.00e-29$, ***$p = 1.08e-52$) and *Mef2cAHFCre;Tbx5*[fl/fl] embryos ($p = 0.93$, $p = 0.81$); *p*-values are based on Mann Whitney *U* test and test whether the spatial distribution of clusters 1 or 2 is more anterior than that of cluster 3. Centre lines show the median, box limits show the first and third quartiles, and whiskers show maximum and minimum values. For statistical tests, the following biological replicas were analyzed: wild-type E8.5: $N = 6$ embryos, $n = 2994$ cells; wild-type E9.5: $N = 6$ embryos, $n = 8618$ cells; *Tbx1*[-/-]: $N = 3$ embryos, $n = 2551$ cells; *Mef2cAHFCre;Tbx5*[fl/fl]: $N = 4$ embryos, $n = 2809$ cells. Scale bars: 50 μm.

account for most of data variance and that is spatially structured, we investigated the impact of loss of T-box gene function on stress patterns in the epithelia. In the presence of TBX5, the *Tbx1*[+] region correlates with cells under isotropic stress and the *Tbx5*[+] region correlates with cells under anisotropic stress and a stress tensor aligned parallel to the antero-posterior axis (Fig. 6). In *Tbx5* conditional mutant embryos, the epithelium is relaxed, as observed from the stress tensor map, and there is an associated loss of cells with anisotropic stress in the posterior DPW (Fig. 6f). As *Tbx5* expression is restricted to the medial region of the posterior DPW, this suggests that *Tbx5* indirectly impacts epithelial tension in a broader region of the epithelium, including the lateral posterior DPW.

The de novo activation of *Tbx5* in the posterior DPW has been shown to be blocked by pharmaceutical approaches using a pan-retinoic acid receptor inverse agonist (BMS493) at E7.5 and E8.5, leading to failure of SHF cell contributions to atrial septal structures at the venous pole[14]. We performed single-cell morphometric analysis of E9.5 mouse embryos exposed to BMS493 after 24 hours of embryo culture. We observed altered stress patterns and cell orientations similar to those seen after genetic ablation of *Tbx5* in the SHF in 2/3 embryos, consistent with loss of cells corresponding to cluster 3 (oriented at 90° in the posterior DPW) and relaxation of the epithelium. One embryo showed a similar phenotype to wild-type embryos (Fig. S12). While consistent with

our findings after conditional *Tbx5* deletion in the SHF, the reduced penetrance possibly reflects heterogeneity in precise embryo stage at the time of BMS493 exposure. Together, these findings point to a previously unidentified role of TBX5 in the acquisition of epithelial tension in the DPW during heart tube elongation.

### Tbx5 regulates actomyosin localization in the SHF
In response to epithelial tension, non-muscle myosin can change its subcellular distribution to form active actomyosin complexes that counteract the forces acting on the cell[49–56]. Diphosphorylated myosin light chain (ppMLC) is the active form of myosin, and its subcellular localization thus provides a read-out of actomyosin activity. At E9.5 the distribution of ppMLC is concentrated on the long apical membrane of elongated cells in the DPW[19]. Here, we show that the accumulation of ppMLC in the SHF coincides with the region where cells from cluster 3 localize, including TBX5 positive cells in the medial posterior DPW (Fig. 7a–a″). At the cellular level ppMLC localizes to the elongated membranes of the cells (Fig. 7b). These results are consistent with previous measurements of the nematic polarity of F-actin and ppMLC[19]. Moreover, this result concurs with the anisotropy maps obtained using force inference revealing an enrichment of anisotropic cells in the posterior DPW. In *Mef2cAHFCre;Tbx5*[fl/fl] mutant embryos, ppMLC accumulation is reduced in the posterior DPW (Fig. 7c–c″). At

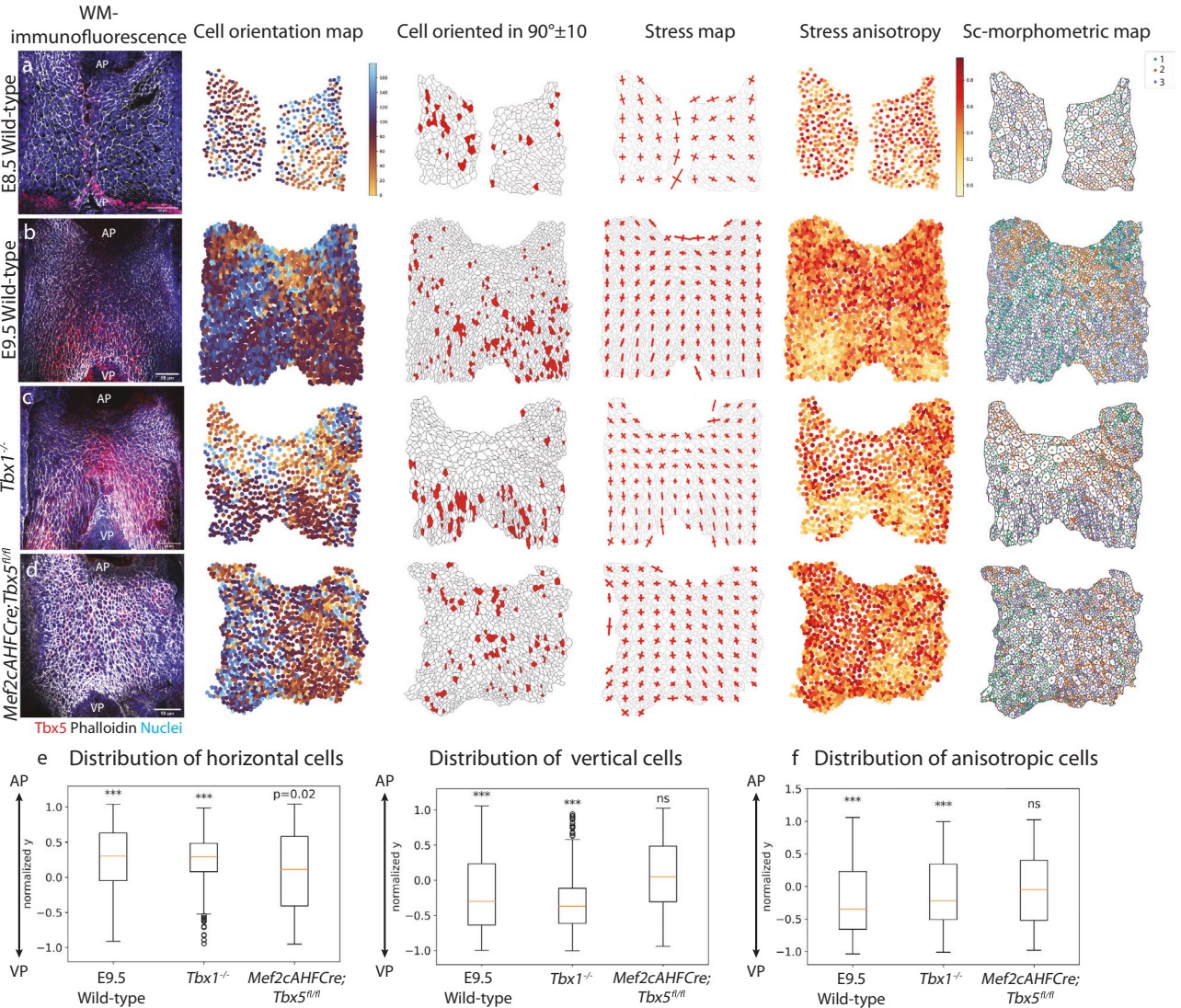

**Fig. 6 | TBX5 induces stress anisotropy and orients cells parallel to the antero-posterior axis in the posterior dorsal pericardial wall. a–d** Whole-mount immunofluorescence of the DPW. Phalloidin in white, TBX5 in red and nuclei in blue. **a–d** Ventral views of the DPW of wild-type embryos at E8.5 (**a**) and E9.5 (**b**) compared to *Tbx1*⁻/⁻ (**c**) and *Mef2cAHFCre;Tbx5*ᶠˡ/ᶠˡ (**d**) mutant embryos, showing from left to right: whole-mount immunofluorescence of the DPW (phalloidin in white, TBX5 in red, nuclei in blue), cell orientation map, spatial plot of the cells whose main orientation angle is 90° ± 10 compared with left-right axis of the embryo on the segmentation mask of the apical cell membrane, cell stress map, cell anisotropy map and morphometric map. **e** Boxplot quantification of the spatial coordinate of cells oriented vertically (90° ± 10) or horizontally (0/180°) along the anterior-posterior axis (with y coordinates normalized from -1 (VP) to 1 (AP)); *p*-values are calculated using a Wilcoxon signed-rank test. For horizontal cells: E9.5 wild-type,

***p* = 1.69e-27, *Tbx1*⁻/⁻, ***p* = 1.17e-18, *Mef2cAHFCre;Tbx5*ᶠˡ/ᶠˡ, p = 0.02. For vertical cells: E9.5 WT: ***p* = 1.25e-39, *Tbx1*⁻/⁻: ***p* = 5.11e-27, *Mef2cAHFCre;Tbx5*ᶠˡ/ᶠˡ: ***p* = 0.96. **f** Boxplot quantification of cells with anisotropic stress (<0.3) along the antero-posterior axis (with y coordinates normalized from −1 (VP) to 1 (AP)). *p*-values are calculated using a Wilcoxon signed-rank test and test whether the spatial distribution of cells deviates from a symmetric distribution. Error bars represent standard deviation. In boxplots in panels **e** and **f**, centre lines show the median, box limits show the first and third quartiles, and whiskers show maximum and minimum values. E9.5 wild-type, ***p* = 5.45e-41, *Tbx1*⁻/⁻, ***p* = 4.34e-5, *Mef2cCreAHF;Tbx5*ᶠˡ/ᶠˡ, p = 0.10. For statistical tests, the following biological replicas were analyzed: wild-type E8.5: N = 6 embryos, n = 2994 cells; wild-type E9.5: N = 6 embryos, n = 8618 cells; *Tbx1*⁻/⁻: N = 3 embryos, n = 2551; *Mef2cAHFCre;Tbx5*ᶠˡ/ᶠˡ: N = 4 embryos, n = 2809 cells. Scale bars: 50 μm.

the cellular level, ppMLC appeared more homogeneously distributed along the cell membranes (Fig. 7d). This altered distribution of ppMLC is consistent with the reduced anisotropy in the posterior DPW of *Tbx5* conditional mutant embryos revealed by force inference. ppMLC localization was also analyzed in embryos exposed to BMS493 in embryo culture. In such embryos (*N* = 4) the polarized pattern of ppMLC distribution that characterizes cells in the posterior dorsal pericardial wall of wild-type embryos was not observed (Fig. S13). Together, these results reinforce the conclusion that in the absence of *Tbx5*, patterned epithelial tension fails to be established in the posterior DPW.

Previous work had indicated that anisotropic ppMLC localization and cell orientation are reduced in the DPW of *Nkx2-5*⁻/⁻ embryos, consistent with lack of epithelial tension[19]. The homeodomain containing transcription factor NKX2-5 is a pleiotropic regulator of heart development required for normal heart tube elongation and patterning and *Nkx2-5*⁻/⁻ embryos die by E9.5 due to cardiovascular defects. Single-cell morphometric analysis revealed loss of cells with anterior-posteriorly oriented stress direction and stress anisotopy in *Nkx2-5*⁻/⁻ embryos (Figs. S14 and S15). This is consistent with a relaxed DPW in *Nkx2-5*⁻/⁻ embryos and with previous observations of loss of cell orientation and homogeneous distribution of ppMLC in *Nkx2-5*⁻/⁻

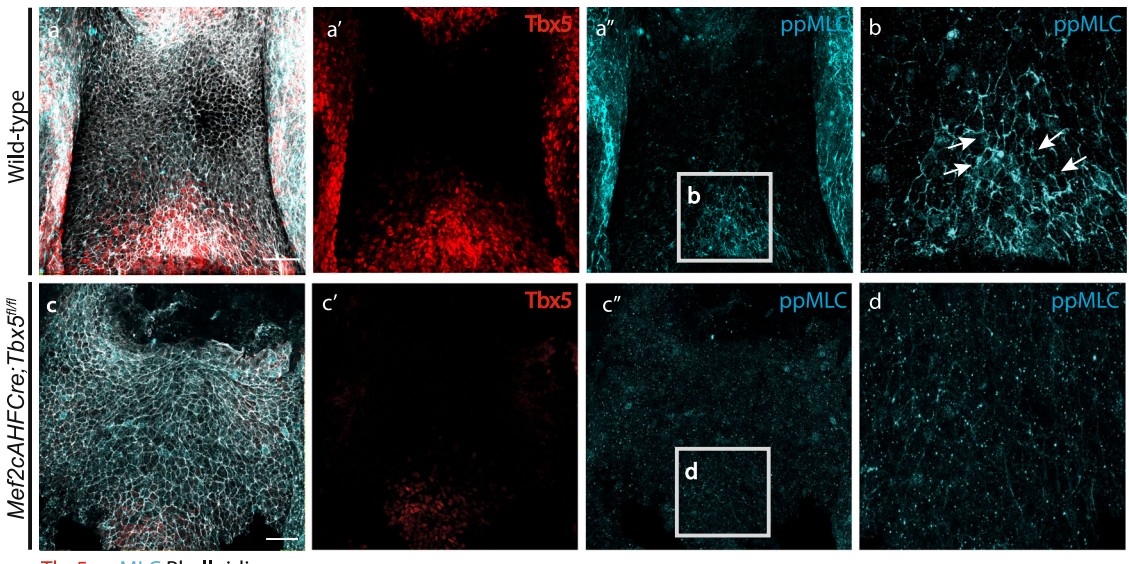

Tbx5 ppMLC Phalloidin

**Fig. 7 | Deletion of Tbx5 disrupts the cellular localization of diphosphorylated myosin light chain in the dorsal pericardial wall. a–a″** Ventral whole-mount views of E9.5 wildtype DPW stained with phalloidin (white), ppMLC (blue) and Tbx5 (red). **b** High magnification view of the posterior DPW in wild-type embryos; white arrows indicate the accumulation of ppMLC at the long membranes of the cells. **c–c″** Ventral whole mount views of E9.5 *Mef2cAHFCre;Tbx5fl/fl* mutants stained with phalloidin (white), ppMLC (blue) and TBX5 (red). **d** High magnification view of the posterior DPW region in *Mef2cAHFCre;Tbx5fl/fl* embryos, showing no accumulation of ppMLC at the long membranes of the cells. Wild-type E9.5: $N = 5$ embryos; *Mef2cAHFCre;Tbx5fl/fl*: $N = 4$ embryos. Scale bars: 50 μm.

embryos at E9.5. Given the similarities with our findings in *Tbx5* conditional mutant embryos, this suggests that SHF addition at the venous pole of the heart, a *Tbx5*-driven process, is a critical determinant in the establishment of epithelial tension in the DPW.

## Discussion

Morphogenesis is the result of dynamic interactions between the chemical and genetic components (or Gene Regulatory Networks)[57] and the physical and mechanical components (or Cell regulatory networks) of developing embryos[37,58]. The embryonic heart is a 3D organ constructed from the folding of epithelial splanchnic mesoderm. Here we have created a quantitative map of the apical morphology and mechanical state of SHF cardiac progenitors in the DPW during heart tube extension and revealed that two T-box genes, *Tbx1* and *Tbx5*, have distinct effects on patterns of epithelial tension in the progenitor field. This provides new insights into understanding the roles of these factors in orchestrating addition of SHF progenitor cells to the arterial and venous poles of the heart.

In the first part of the study, we characterized the shape and mechanics of SHF cells throughout the DPW. We established a pipeline to perform single-cell morphometrics in SHF progenitor cells unifying geometrical measurements of the cells and mechanical measurements using recent image analysis tools, in particular force inference[31–33]. The pipeline has helped to pinpoint the main features clustering the cells in the epithelium: orientation and stress direction. This result highlights the importance of cell mechanics during heart morphogenesis. These forces may arise through progenitor cell addition to the growing heart, initially across the dorsal mesocardium and later at the poles of the heart. Previous work has suggested that a mechanical feedback loop between progenitor cell deployment and epithelial tension in the SHF acts as a motor driving heart tube elongation[18,19]. The patterns of epithelial tension observed in our single-cell morphometric analysis provide further support for such a model. Moreover, our findings suggest that establishment of opposing directionality in trajectories from the SHF to the heart through activation of a *Tbx5*-dependent venous pole progenitor cell program is a critical step in driving epithelial tension in the DPW. Our data also show that nucleus-to-Golgi polarity is among the least important features in the classification of

the cells; indeed, the apical position of the Golgi highlights the apicobasal polarity of SHF cells[17]. This strengthens the idea that changes in cell shape and stress in the DPW epithelium are a passive response to external forces rather than being driven by active intrinsic forces, congruent with SHF deployment to the cardiac poles within an apicobasally polarized epithelial sheet.

The single-cell morphometric pipeline has allowed us to map SHF progenitor cells in the DPW according to their morphological and mechanical properties. Once the two cardiac poles have been established at E9.5, our map reveals bilaterally symmetric clusters alternating around the arterial pole. This result is consistent with previous studies showing rotation of the elongating outflow tract wall, ultimately ensuring correct ventricular arterial alignment[59]. Moreover, genetic studies have revealed differences in gene expression in left and right sides of the DPW, including the asymmetric expression of *Nodal*, *Pitx2c* and *Lefty* in left sided SHF progenitor cells[60,61].

We identified a third cluster labeling cells in the posterior DPW. This cluster represents a subpopulation of progenitor cells that are oriented and anisotropically stressed at 90°, along the antero-posterior axis. This spatial clustering correlates with the establishment of distinct arterial and venous pole programs within the SHF and provides a read-out of cell properties as cells are added to the opposite poles of the heart, cells in the anterior part (clusters 1 and 2) being added to the arterial pole and TBX5 positive cells in the posterior domain (within cluster 3) being added to the venous pole. Our results thus highlight a symmetry breaking step in the DPW after dorsal mesocardial breakdown as *Tbx5* activation in the posterior DPW establishes a venous pole progenitor cell population with distinct epithelial properties. Indeed, comparison of single-cell morphometric maps of the DPW of E8.5 and E9.5 wild-type embryos as well as *Tbx1−/−* and *Mef2cAHFCre;Tbx5fl/fl* mutant embryos, revealed the *Tbx5* dependence of the regionalization of cluster 3 at the venous pole. In the absence of *Tbx5*, either at E8.5, before *Tbx5* expression in the DPW, or through its genetic disruption in *Tbx5* conditional mutant embryos, cluster 3 is not regionalized. TBX5 thus appears necessary to establish the 3rd posterior morphological cluster and to drive epithelial tension and cell orientation parallel to the antero-posterior axis in the posterior DPW. Consistent with this conclusion, symmetrizing cell

orientation and stress direction allowed cells be classified into two clusters distributed anteriorly and posteriorly at E9.5 but not E8.5. Moreover, loss of regionalized distribution of these clusters in *Mef2cAHFCre;Tbx5*<sup>fl/fl</sup> embryos points to at least two independent forces driving epithelial stress in the DPW: TBX5-dependent patterned tension close to the venous pole and TBX5-independent stress as cells converge bilaterally on the arterial pole.

Genetic perturbation experiments show that TBX1 is important to maintain the maximal cell number within the epithelium, in agreement with previous studies[10,15–17]. We observe a loss of the two main bilateral populations in the aDPW; however, the loss of TBX1 does not affect the establishment of epithelial stress in the posterior SHF. On the other hand, we demonstrate that TBX5 is required in the SHF lineage to orient cells in the posterior DPW parallel to the antero-posterior axis of the embryo at E9.5 and to establish anisotropic stress. This result suggests that the activation of *Tbx5* expression in the SHF cells may directly modulate the mechanical properties of cells in the posterior medial DPW. For example, this may be through regulating ppMLC localization to the elongated membranes of the cells that accompanies cell orientation in 90° in the posterior SHF. Interestingly, YAP and TAZ have been shown to directly interact with TBX5 to regulate transcription; future experiments will investigate whether they interact with TBX5 in the posterior DPW to regulate SHF deployment[62,63]. However, as the orientation and tension pattern that defines cluster 3 is observed in a broader posterior DPW territory than the TBX5+ domain, TBX5 may also or alternatively play an indirect role by exerting forces on the epithelium through TBX5-dependent SHF deployment to the venous pole of the heart. Similarly, loss of NKX2-5, associated with failure of heart tube extension at both poles, leads to loss of posterior enrichment of cells from cluster 3 and relaxation of the progenitor cell epithelium. Together these observations suggest that the onset of SHF cell deployment to the venous pole may drive the establishment of patterned tension in the DPW during heart tube extension. We suggest that the establishment of a distinct TBX5-dependent pathway for cell addition from the DPW to the elongating heart tube through the venous pole would create an opposing force to an earlier established arterial pole pathway, placing the epithelium under mechanical tension.

In early E9.5 embryos (<20 somites) cells from cluster 3 can be observed in the anterior medial region while only in later E9.5 embryos (>20 somites) is the 3rd cluster is restricted to the posterior region. This result provides insights into changing apical cell morphology of cells in the DPW during the dynamic process of heart tube formation. Of note, the DPW is in continuity across the dorsal mesocardium with the *Tbx5* expressing FHF-derived heart tube at early somite stages[14]. TBX5 may be involved in cell shape changes taking place during dorsal mesocardium breakdown as well as in the later posterior DPW.

With this study, we have created a morphometric map of the SHF cells before and after dorsal mesocardium breakdown and have shown that T-box genes regulating addition of the progenitor cells to the heart tube affect the mechanical state of the DPW epithelium. This provides an example of how links between genetics and mechanical forces may coordinate organogenesis. Future perspectives include coupling single-cell morphometric analysis with single-cell RNA-seq analysis to generate an integrated map of transcriptional and morphological cell features during SHF deployment. These perspectives have begun to be explored in the mouse brain, where mechanical patterns have been examined in brain compartments with different transcriptomic profiles[64].

Limitations of our study include the lack of tools to perform live imaging. The data analyzed in this work represent temporal snapshots of DPW development during SHF deployment and intermediate stages have not been addressed. There are significant challenges in dynamically imaging the DPW, which at these stages is located in the central part of the developing embryo and is currently not accessible to imaging over time even using light sheet microscopy. In addition, the single-cell morphometric analysis performed in this study has focused on the apical side of the epithelium. The DPW is an atypical epithelium as the cells are epithelial-like apically yet have dynamic mesenchymal-like features baso-laterally[17]. This particular cellular conformation, together with the fact that active actomyosin accumulates on the apical side of cells in the DPW suggest that apical morphometrics provide an accurate readout of epithelial tension. However, exchanges with the extracellular matrix (ECM) baso-laterally, not evaluated here, and or adjacent cell types such as foregut endoderm, may also impact on overall epithelial morphology in the DPW.

Finally, we propose an unsupervised pipeline to delineate regions in space with similar morphometric and mechanical features. This pipeline starts with a dimensionality reduction step that selects the features which, on top of explaining most of the variance, as in classical PCA, also satisfy the criterium of displaying spatial regionalization. This procedure can be extended to different embryos of the same condition, using joint diagonalization, a step termed joint spatial PCA. The pipeline developed here for analysis of progenitor cells based on image analysis tools and unsupervised learning approaches can be used to analyze single-cell measurements with any currently available segmentation and quantification tool.

## Methods

### Mouse strains and embryo generation
The following mouse lines were used: *Tbx1*<sup>+/-</sup>[7], *Mef2c-AHF-Cre*[48], *Tbx5*<sup>fl/fl</sup>[9], *Nkx2.5*<sup>+/-</sup>[65] and CD1 mice. Mice were maintained on a mixed C57Bl/6 and CD1 background and genotyped by PCR using standard protocols from DNA extracted from hair, tail tips, or yolk sacs with primers listed in Supplementary Table 1. Embryos were collected at embryonic days (E) 8.5 or 9.5, dated from noon of the day of the vaginal plug (E0.5). Embryos were not genotyped for sex. Animal experiments were carried out in agreement with national and European law and approved by the Ethics Committee for Animal Experimentation of Marseille and the French Ministry for National Education, Higher Education and Research (Apafis No. 10266-2017061618121519).

### Embryo dissection and whole-mount immunostaining
Embryos were collected and fixed for 1 h in 4% paraformaldehyde (PFA) for routine whole-mount immunostaining. After fixation, the heart was removed to expose the dorsal pericardial wall and embryos processed for whole mount staining. For immunostaining, embryos were blocked overnight in blocking buffer (PBS/0,1% Triton/3% BSA), incubated overnight at 4 °C with primary antibodies, washed in blocking buffer and incubated overnight at 4 °C with species-specific secondary antibodies. Finally, embryos were stained for F-actin with phalloidin and counterstained with Hoechst. After staining, the pharyngeal region was microdissected and the dorsal pericardial wall mounted and imaged ventrally using a confocal LSM 780 or 880 microscope. Confocal images were acquired as Z-stacks. The following antibodies were used: goat anti-TBX5 (1/250, Santa Cruz sc-17866), rabbit anti-Cre (1/500, Novagen 69050), rabbit anti-ppMLC (1/200, Cell Signaling Thr18/Ser19 3674), mouse anti-Golgi (GM-130) (1/100, BD Transduction Laboratories 610823). Fluorescent secondary antibodies Alexa 488, 568 and 647 from Jackson ImmunoResearch and Invitrogen were used at 1/200. Two fluorescent forms of phalloidin were used to label F-actin: phalloidin 647 (1/50, Sigma 65906) or phalloidin 488 (1/50, Sigma 49409). In this work, we included single-cell morphometric analysis of images from a previous study[19] for 2 *Tbx1*<sup>-/-</sup> embryos; 4 *Nkx2.5*<sup>-/-</sup> embryos; and 3 E8.5 wild-type embryos.

### Mouse embryo culture
E8.5 embryos were dissected in $CO_2$ independent medium (Invitrogen, 18045-054), 0.04% BSA (Albumin, from bovine serum, SIGMA A88056), 1% penicillin/streptomycin (Invitrogen, 15140-122) without damaging

the yolk sac. Embryos were cultured in rolling-bottles for 24 h in 75% rat serum (Harlan SR-0100 or IGBMC Strasbourg), 25% DMEM (Invitrogen), and 1% penicillin/streptomycin. For the first 12 h of culture embryos were exposed to a mixture of 5% $CO_2$, 5% $O_2$, 90% $N_2$ and subsequently switched to 5% $CO_2$, 20% $O_2$, 75% $N_2$. For pharmacological treatment, control embryos were cultured with DMSO at 1 μl/ml and treated embryos with the panRAR antagonist BMS493 at $10^{-5}$ M (Bristol-Myers-Squibb, Princeton, NJ, USA; B6688 SIGMA). After culture, embryos were removed from the yolk sac and fixed for 1 h in 4% paraformaldehyde prior to whole-mount immunostaining.

## RNAscope on whole-mount embryos

Embryos were fixed overnight in 4% paraformaldehyde (PFA) and then dehydrated in methanol. After fixation, the heart was removed to expose the DPW. Whole-mount RNA-FISH was performed according to the protocol of the RNAscope Multiplex Fluorescent v2 Assay (Acbio; cat. No323110)[66]. Embryos were imaged using a confocal LSM 780 or 880 microscope. The following probes were used: mm-Tbx1-C1 (cat no.481911), mm-Tbx5-C2 (cat no. 519581). For pixel intensity quantification, a ROI line in half of the left part of the embryos was drawn, creating a sagittal section from arterial pole to the venous pole. For each mRNA channel the intensity was measured along the ROI. The measurements were normalized using the min-max-normalization, scaling the values in the array range from 0 to 1. The profile plot of the measurements was visualized using ImageJ. The algorithm uniform_filtered1d from the scipy library was used to smooth the profile plot.

## Confocal microscopy, image analysis, and quantification

The DPW was imaged ventrally as a Z-stack using a Zeiss LSM 780 or 880 confocal microscope with a 40X objective. The 20X objective was used to have an overview of the full embryo stage. The apical surface of the DPW epithelium was imaged in a ventral view. The apical cortical F-actin belt was used as an apical membrane marker for segmentation. Max intensity projection and background noise cleaning per channel was performed prior to segmentation using LocalZprojector (LZP) imageJ plug-in[31]. Segmentation of the epithelium was performed using Tissue Analyser software[32]. The software is an ImageJ plugin using a watershed algorithm to segment the cell cortex. Segmentation was performed on the entire DPW. Briefly, base segmentation was automatically performed by the Tissue Analyser software and corrected manually. Apical surface area, perimeter, eccentricity or apical circularity index (0 means that the cell fits a rounded shape, 1 means that the cell fits an ellipse shape), cell orientation, and number of neighbors were generated using Deproj MATLAB plug-in[31]. Binned cell elongation axis plots were generated using Tissue Analyser. Segmentation data was stored in the form of a.xlsx file. Stress measurements including cell pressures, stress main direction ($\vartheta_1$) and stress anisotropy, which is the ratio between the amplitudes of the stress tensor ($\lambda_1/\lambda_2$) were calculated using force inference MATLAB plug-in[33]. Force inference curvature tests were performed to calculate the radius of tissue and cell curvature (Fig. S16). In specific cases (Figs. 2k, S4b, S5, 6, 9, and 10), we have computed our pipeline on AP-symmetrized orientation and stress direction. AP-symmetrized means that we consider the absolute value of the (orientation – 90°) and the absolute value of the stress direction, making these two features symmetric with respect to the antero-posterior axis. Stress maps and pressure maps were generated using the same plug-in. Stress anisotropy maps were generated using a custom-made Python code: if the ratio is close to 1, the cells are under isotropic stress and are color coded in red, if the ratio is close to 0, the cells are under anisotropic stress and are color coded in yellow. Morphological and spatial data were stored together with mechanical data in the form of a xlsx and csv file for each embryo.

## Data analysis for single-cell morphometrics

The pipeline is comprised of two steps. The first step, called joint spatial PCA, projects the cells along the axis that maximizes the variance in the data while at the same time enhancing features that are spatially regionalized. The second step is training a Gaussian model on the projected data points to distinguish clusters, that are then mapped back onto physical space to reveal their location. The pipeline is applied to the E9.5 WT embryos. Then, the cells from other conditions embryos are projected on the same axis, and the Gaussian model trained on the E9.5 WT data is used to cluster the additional embryos. The input for the training on E9.5 WT embryos is the respective adjacency matrices (n x n) that express the neighborhoods of cells in space and the features matrices ($n \times 8$) where $n$ in the number of cells in the embryo. Each column of the features matrices are centered and scaled for each embryo separately. Then, for each embryo, the matrix M_k (k = 1:8) is expressed as $(X^t(L+L^t)X)/2n$ where $L$ is the row normalized adjacency matrix. This matrix M_k is analogous to the covariance matrix in classical PCA. The difference is that the eigenvectors associated with highest eigenvalues not only explain most of the variance but also display high spatial auto-correlation[36]. These matrices are then jointly diagonalized by a function made available by Cardoso et al.[42] in MATLAB. This consists in finding the optimal common eigenvectors for the set of matrices that approximate their diagonalizations. We rank the eigenvalues and select the highest, and take its associated eigenvector as projection axis jsPC1 (Fig. 1d,e). We project all the datapoints on this axis. The main features contributing to this axis are cell orientation and cell main stress direction. Then, we fit a Gaussian mixture model on all cells from embryos of E9.5 WT condition projected on the same axis jsPC1. The Akaike score gives 3 as the optimal number of clusters (Fig. S4). The cells from other conditions, all centered and scaled individually, are then clustered using the parameters of the previously fitted Gaussian mixture model learnt on E9.5 WT embryos. The different steps of the pipeline are implemented in Python, except for the function for approximate joint diagonalization implemented in MATLAB. Additional cluster maps showing each cluster independently are shown in Figs. S17–S20.

## Statistics and reproducibility

Segmentation and analysis was performed on three or more embryos of each of the different developmental stages, genotypes or manipulations, as follows (N is the number of embryos, n the number of cells): E8.5 wild-type embryos, $N = 6$, $n = 2996$; E9.5 wild-type embryos, $N = 6$, $n = 8618$; *Tbx1*$^{-/-}$ embryos, $N = 3$, $n = 2551$; *Mef2cAHFCre;Tbx5*$^{fl/fl}$ embryos, $N = 3$, $n = 3695$; *Nkx2.5*$^{-/-}$ $N = 4$, $n = 5568$; BMS treated embryos, $N = 3$, $n = 3432$. All data are included in the manuscript and Supplementary Figs. The results of single-cell morphometrics were obtained by pooling the 6 E9.5 wild-type embryos. The results were used to predict the cells labeled in the different temporal and genetically modified embryos. For cluster signatures the means were calculated and plotted in the distribution of the values for each feature and cluster number. Statistical analysis of mean values for antero-posterior and left-right distance was performed using two-tailed $T$-test. Mann Whitney U test from scipy.stats.mannwhitneyu was used to calculate the p-values in cluster boxplots (Fig. 5). Wilcoxon signed-rank test from scipy.stats.wilcoxon was used for vertical (> 90°), horizontal (<10°) boxplots and anisotropic cells (<0.3) in Fig. 6. For all boxplots, the y-positions were normalized as follows: $n_{ormalized} = -2(y-<y>)/(Y_{max}- Y_{min})$; ns, non-significant; **$P < 0.01$; ***$P < 0.001$. Error bars represent standard deviation (SD).

## Reporting summary

Further information on research design is available in the Nature Portfolio Reporting Summary linked to this article.

## Data availability

Source data are provided with this paper as a Source Data file and available on the Github: sc-morphometrics-SHF. Source data are provided with this paper.

## Code availability

The measurement of Golgi-to-nucleus polarity in 3D and the sc-morphology analysis is available on Github sc-morphometrics-SHF [https://github.com/VILLOUTREIXLab/sc-morphometrics-SHF]. The code of the Bayesian force inference has been published[33]. The Joint Spatial PCA is available on the Github JointSpatialPCA [https://github.com/VILLOUTREIXLab/JointSpatialPCA].

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

## Acknowledgements

We are grateful to our colleagues in the Kelly and Villoutreix labs for helpful discussions. The project leading to this publication has received funding from France 2030, the French Government program managed by the French National Research Agency (ANR-16-CONV-0001) and from the Excellence Initiative of Aix-Marseille University A*MIDEX (Turing Centre for Living Systems (P.V. and C.G.). We are grateful to the Agence National pour la Recherche (Heartbox ANR-18-CE13-011 and Heartbound ANR-22-CE13 projects (R.G.K.), Fondation pour la Recherche Médicale DEQ20150331717 (R.G.K.) and FDT202304016591 (C.G.), AFM-Telethon and Fondation Leducq Transatlantic Network of Excellence 15CVD01 (R.G.K.) for financial support and the France-Biolmaging/PICsL infrastructure (ANR-10-INSB-04-01).

## Author contributions

C.G., P.V., and R.G.K. designed experiments. Experiments were performed by C.G., S.S., B.A., and R.C. to develop codes for data analysis and visualization. C.G. and S.S. performed the analysis and quantifications. C.G., S.S., R.G.K., and P.V. wrote the manuscript. All authors provided intellectual input and read and approved the manuscript. R.G.K. and P.V. managed the project.

## Competing interests

The authors declare no competing interests.
