## [Peer Review file · Nature Communications]

Single-cell morphometrics reveals T-box dependent patterns of epithelial tension in the Second Heart field

Corresponding Author: Dr Robert Kelly

Version 0:

Reviewer comments:

Reviewer #1

(Remarks to the Author)

The manuscript by Guijarro et al explores the morphology of the apical side of the cells of the dorsal pericardial wall at early stages of heart tube formation in the mouse. They approach this problem by imaging and segmenting individual cells, which allows them to classify cells in an unsupervised manner, according to a selected set of geometrical parameters. Using this strategy, they classify groups of cells according to the most informative parameters for cell classification and report that cells close to the arterial pole present different morphologies and orientation than those close to the venous pole. Further to this, they study the expression of Tbx1 and Tbx5 in this context and characterize the changes in apical cell morphology in Tbx1 and Tbx5 mutants. While the approach undertaken in this manuscript is very attractive and potentially informative about developmental mechanisms, there are also important limitations that question the relevance of the observations. Overall, the conclusions on the mechanisms that govern heart tube morphogenesis are rather limited.

Main points:

- The study is based on acquiring a 2D projection of a 3D specimen, which distorts the real size and shape of the apical surface of cells. A 3D acquisition followed by a cartographic methodology to render realistic 2D apical surfaces would be required in this study. This is particularly important at the arterial and venous poles, where the dorsal pericardial wall cell sheet bends to adapt to the heart tube, but also at lateral sides of the specimens.
- The study is based on the characteristics of the apical surface of the cells and does not consider the 3D geometry of the cells or the interactions of the cells with the basement membrane. Based on several previous studies and the known behavior of epithelia, the exclusion of this aspect limits the relevance of the study.
- The study claims the need of "Machine Learning" approaches to solve the problem of understanding "Morphogenesis". Furthermore, they claim to have applied "Machine Learning" approaches and to have developed a software pipeline for the application of this "Machine Learning" approach. The reality is that the manuscript does not describe or apply any new "Machine Learning" approach but the use of well-established classification algorithms such as K-means. This is a very limited application of "Machine Learning" procedures.
- The parameters used for cell classification are few and rather redundant, such that in the end, the authors use nearly a unique parameter for cell classification.
- The use of cell morphology to generate a map of forces is only an inference, but true forces or tensions are not measured. The limitation of this approach adds to the lack of characterization of the 3D structure of the cells and of their basal-side interactions.
- The alternance of cell group identities around the arterial pole seems artifactual and derived from the bilateral symmetry or radial symmetry of the cell disposition around the outflow tract insertion site. The conclusions made about arterial pole rotation based on this classification are misleading. The authors measure angles with respect to a L-R line and this scores cells oriented symmetrically with respect to the midline (or outflow tract insertion circumference) into different cell clusters, when, in fact, they are similarly oriented with respect to the embryonic symmetry references. The absolute values of angles with respect to the midline or to the closer tangent to the outflow insertion circumference should be used instead to determine the true classification.

Other points

- The study ignores proliferation of the cells under study. Related to this, what does group 4 from the K-means classification represent? Could this group represent dividing cells, considering their widespread distribution? This would be an interesting

aspect of the study.

- The procedure to “clean” the distribution of cells belonging to the different clusters (Figure 2d) provides nicer images by neglecting the real data, but it is not justified by any biological argument.
- Line 335: “In Mef2cAHFCre; Tbx5fl/fl embryos cluster 3, distinguishing posterior DPW cells in wild-type embryos, is not observed (Fig. 5c)”. Actually, in Figure 5c’ a numerous group-3 cells can be observed.
- Line 363: “it can be observed that in presence of Tbx5, there is an enrichment of horizontal cells in the anterior DPW and an enrichment of cells oriented vertically in the posterior DPW (Fig. S9e).” Here, the authors are describing the wild type situation. Using “enrichment” to describe the wild type specimens is misleading.
- Calls in the text to Figure 1 do not match Figure panels

Reviewer #2

(Remarks to the Author)

This manuscript investigates the interplay between genetic factors and mechanical forces during heart development. It focuses on how progenitor cells from the second heart field (SHF) contribute to heart tube extension. Utilizing single-cell analysis, the researchers explore how cell shape and mechanical stress are related. The aim of this study is to generate a quantitative map of cell shape and mechanical features in of SHF progenitor cells and to study how different T-box genes, Tbx1 and Tbx5, affect these features.

The authors not only generate such a feature map of cells but find that Tbx1 guides cell orientation towards the arterial pole. On the other hand, Tbx5 activation aligns cell morphologies. These insights underscore the intricate connections between genetics and mechanical aspects in cardiac morphogenesis. The findings can have a significant impact on understanding not only heart development, but congenital heart defects.

Overall, the underlying idea of this study can have a significant impact on the field of heart morphogenesis and Evo-Devo in general. While for other biological systems, morphometric studies have already been done (e.g., Andrews et al. *Development* 2021 [27] or Viana et al., *Nature* 2023), to my knowledge there are no published studies that also consider force inference features, except a very relevant preprint by Hallou *bioRxiv* 2023. Hence, including mechanical features for a morphometric analysis is novel and a very promising idea for future studies of biological systems.

The idea behind the methodology of this study is in general sound but requires improvement (as pointed out below in the list of comments). My main concern is that it is not clear if advanced machine learning tools are really required for this data set that only uses 9 features and what value the PCAs/UMAPs provide.

Moreover, In its current form the manuscript lacks some explanation or at least a biological/biophysical hypothesis for the observed behaviour that goes beyond providing a map of cell features in the developing heart.

Below you will find comments that will hopefully help to improve this manuscript.

Major Comments:

- Fig 1f-i and Fig. 2a-d proof that the cells are distributed in a continuum, disproving the existence of distinct clusters/populations (see also text line 198 for the PCA and text line 201-202 for the UMAP where this exactly is pointed out). What exactly are the clusters/populations that the authors find then? It might be better to treat it as a continuum and not as N distinct clusters.
- In Fig. S1a it seems like the tissue is curved. Is that taken into consideration when extracting the shape features and mechanics?
- Did the authors check that the 9 features are not correlated? Fig 1d seems to show that for example area & perimeter, cell orientation & stress direction and maybe cell neighbours & Golgi polarity are strongly correlated and usually one would do a PCA/UMAP with only one of them. Also, if one only has 6 to 9 parameters, it is not clear why dimensional reduction techniques are required at all. Two possible solutions I see are to extend the features studied to provide a more holistic description of the cell properties. Or reduce the number as features that are relevant as much as possible and do not perform the dimensional reduction.
- It is unclear to me why the authors perform the UMAP and dimensional reduction in general. The main findings about the different cell populations/clusters are mostly disconnected from the data. For example, what insights does Figure 5f provide? Please also note this recent paper (Chari, Pachter, *Plos Comp Biology* 2023) which shows that UMAP is not a good method for cluster verification.
- Clustering using the k-means algorithm can be improved. My concerns are that k-means in some special circumstances can bias towards equally sized clusters. Can the authors verify their results using other clustering algorithms? Also, when performing the k-means clustering, how do the authors define the initial points of the k-means clusters. How many initial conditions did the authors try out and how robust is the calculation of the ideal cluster number?
- Finding the individual clusters 1, 2 and 3 is one of the main highlights of the authors manuscript. My main concern is that these clusters are apparently defined several times and for different underlying data set (e.g., for Fig. 2 the authors use E9.5 embryos, for Fig. 24 the authors use different time points, for Fig. 5 the authors use mutants). But how can the authors compare the different clusters if each time the authors perform a different k-means clustering? Shouldn't the authors only define the clusters once for the WT and then classify the cells in the other conditions using the WT training data? For example, Fig. 2e-h and Fig. 5e show qualitatively very similar distributions. Additionally, it would be helpful if the authors could give the clusters specific names and call them “xyz population”. It is confusing to call them clusters that in Fig. 3d’ form also distinct spatial clusters. Better to say that cells of population xyz (the orange ones) form two distinct clusters in Fig. 1d’’. Also, often it is not clear which data was used to define the so-called

cluster 3, e.g., in line 354.

- If the authors need to consider different conditions for the clustering or dimensional reduction of cells, how do they weight the different cases? E.g., if the authors compare 1000 cells of the WT and only 100 cells of mutant tissue, shouldn't one weight the data accordingly so that each condition contributes equally?
- Is it possible and feasible to make proper mechanics measurements (e.g., laser ablation experiments) to check if the force inference results/stress maps are correct?

Other Comments:

- Line 41: It might be good to say what Tbx1 is in the abstract.
- Line 91 and 430: The concept of CRNs is not really established and needs more explanation in the manuscript. In general, CRNs as a concept seem to be badly named since genes and gene expression are part of cell biology and one would assume that GRNs are just a sub-group of CRNs.
- Line 130: of course, image analysis is non-invasive. I guess the authors mean non-invasive force measurements based on image analysis. Also, would the authors call the imaging/microscopy they perform non-invasive?
- Line 169: the number of cell neighbours is not a morphological feature!
- Fig. 2e-h and Fig. 5e: Unclear how the distributions are normalised. Clearly, the area is not the same for all curves. Best is to show the probability density function.
- Line 170: The angle between the centre of the nuclei to the centre of the Golgi, relative to the embryonic left-right axis goes from 0 to 360. Are these also the numerical features the authors use for the analysis? If yes, that would mean that the numerical features are very different for 1 and 359, while the directions are very similar?
- Line 200/201: Fig S2 is supposed to show reproducible behaviour among all embryos, but indeed does not show anything in this direction. But in fact, it would be interesting to show the morphospace and feature distributions for all individual embryos.
- Line 223: Why exactly do the authors use a UMAP and not PCA?
- Line 225: I understand that the authors want to reduce imaging noise. But why would the authors want to reduce biological noise, while this could also provide useful insights? It might be interesting to look at biological noise.
- Line 231-236: Is the ideal cluster number 3 for all individual embryos?
- Line 241-244: How do the authors find these angles? Are these the exact numbers extracted from the data?
- Fig. 4: Can you show each channel individually? Right now, the composite image is very hard to see any co-expression. Overall, I have problems linking text lines 295-305 to Fig. 4 and it would be good if the figure can be improved so that it is easier to understand in which regions exactly to look.
- Is it possible for the same cells to look at shape, mechanics and Tbx1 and Tbx5 expression? If that is experimentally feasible, it would be interesting to quantify individual gene expression of cells.
- Line 329: I am not sure how I am supposed to see this from Fig. 5f alone.
- Line 334-335: The data does not seem to show that the percentages are the same as in the WT. Could you show that the observed difference in percentages is not significant?
- Fig 6 and from line 351: Could the authors show in the figure where the specific clusters are to make it easier to follow the text?
- Line 382 – Line 389: Could this be explained by a too low concentration of BMS493?
- Line 430 – 431: Turing did not consider mechanics in his theory of mechanics, but interactions of different biochemical cues with different diffusion coefficients. On the other side D'Arcy considered the role of mechanics on morphogenesis, but with no focus on biochemistry.
- Line 441 and 485: I am not sure that it is appropriate that the authors addressed the regulatory networks. They did not analyse the feedback or proper interactions between the individual features.
- Figure 5a'-c': Why do the authors not show their connectivity graphs here with the "cleaned" cluster distributions?
- Fig. 6a" and similar figures: The angles are continuously distributed. Hence, the probability to find exactly an angle of 90 is zero and the authors need to define a range they consider defining the red cells. What is this condition?

Typos and small errors:

- Fig. 1b and other figures: Color-coding cell orientation and cell eccentricity like the cell area might look better. Additionally, some figures have a very low resolution (e.g. Golgi figure in Fig. 1b). But this might have happened during the submission and does not need to be the fault of the authors.
- Fig. 1b and 1c and all other figures: colorbars need units. Also, in Fig. 1c, the cell pressure has no colorbar. Another example is Fig. 1e where it is not shown what the the y-axis is.
- Line 121-122: provide a reference.
- Line 192: Do the authors mean Fig 1D?
- Line 198-199: What is X and Y?
- Line 199: I can't find Fig. 1k.
- Line 201: Unclear what is meant by "Linear distribution"? A linear distribution
- Line 203-208: This text is written very unclear. What does it mean that cell elongation is different on the left and the right side? And what are gradients according to apical area, perimeter, and pressure? Also, Fig 1h'-j does not show what is written in the text and 1j does not exist.
- Sometimes figures do not appear in the right order, e.g., Fig. 3d'" (line 268) is shown before 3d". I suggest the authors go again over their figures and the text and see how they can improve the synchronicity between them.
- Line 270-271: This sentence is not clear to me. Where is this shown?
- Line 272: What means above?
- Line 433: What is the final architecture of an embryo? An embryo is a transitional configuration of cells and to a (probably very limited degree) the adult is the final architecture, not?
- Line 465: typo "including"

- Line 476: typo “deponedence”
- Line 476-Line 479: This sentence is very hard to comprehend.
- Line 541-542: How is the connectivity graph a mathematical representation of a CRN? It is unclear to me what the CRN is in that case? Is it a network of the different features or different cells?
- Line 594-595: Isn't a circle the same as an ellipse with identical short and long axis? Then how can 0 be a rounded shape (circle?) and 1 be an ellipse?
- Line 608: Before the authors said it was xlsx files, what are the csv files?
- Fig. S2: typo “Dimentionality”
- Fig. S3a: Why are there two cell areas?
- Fig. S6e: Figure seems to be cut.

Version 1:

Reviewer comments:

Reviewer #1

(Remarks to the Author)

The authors have improved the methodology and answered several of my concerns, however some issues still remain to be addressed:

1.- In my opinion, the results of the deformation/tension maps indicate that there are two colliding stresses; one tangential to the outflow track insertion circumference and another in the A-P direction. The A-P influence is predominant posteriorly, but it is not exclusive of this region, given that group-3 cells can be seen as well in anterior regions. And vice versa, groups 1 and 2 also show posteriorly located cells intermingled with group-3 cells. The outflow track influence is obviously more predominant anteriorly. In my opinion, the way clusters are defined and explained in the manuscript fails to explain this organization. In particular, the maintenance of two groups for the anterior region is particularly problematic as this is derived from measuring cell deformation in the A-P direction, whereas the organization of the tissue in this region follows a radial symmetry. The maintenance of these two anterior groups may mislead the reader into thinking that there is a L-R difference. In my opinion, the subdivision in two clusters shown in Figure S4b is cleaner and much more descriptive of the biology of the system. Readers would benefit from seeing the changes that take place in these two clusters in Tbx1 and Tbx5 mutants.

2.- The establishment of a particular threshold to determine that anisotropy does not exist in a cell is arbitrary and the chosen value (0.3) appears too low to consider that measurements above this threshold are isotropic. Apparently, this threshold is set at 0.08 for figure 6 (See line 656 and Figure legend) but I think this is a mistake, given that it is set at 0.3 in Figure S5 and all the rest of Figures with the same type of representation. In any case, it would be better to show a continuous color scale reflecting all measurements and their distribution in the tissue for all specimens (at E8.5 and E9.5). This would also show the anisotropy map independently of the stress direction.

3.- It would be important for readers to see the distribution of each cell cluster in every specimen, as shown in Figure 2b for a single specimen.

4.- Authors should explain better what was measured and statistically analyzed in Figure 6e and 6f

5.- In Figure S1, the authors describe their 3D analysis of the golgi-to-nucleus orientation measurements, however, they do not specify the thresholds used to determine the polarity. Unfortunately, the color scale cannot be properly visualized in the graph shown in Figure S1D (below) and this should be improved

Reviewer #2

(Remarks to the Author)

This manuscript presents a comprehensive analysis of the role of T-box genes in regulating epithelial tension during cardiac morphogenesis. The study utilizes advanced single-cell morphometrics and force inference techniques to elucidate the dynamic interplay between genetic and mechanical factors in heart tube extension. The findings offer significant insights into the molecular mechanisms underlying heart development and congenital heart defects.

The authors have made significant improvements to the previous version of the draft. Especially using the spatial PCA seems very appropriate and provides useful insights into studying SHF.

I believe this manuscript can provide significant contributions to the study of heart development and their methodology will also be very useful for studying other 2D tissues. Yet, I have some concerns that need to be accounted for before I can give my final recommendation for this paper:

- The author's reply to Reviewer 2 (major comment 2) and the methods section imply that the force inference was done on depth-corrected data. However, Figure 1a does not really illustrate that the force inference was applied to the depth-corrected data. Would the depth correction not lead to distortions that will also affect the inferred force field? I expect that the curvature is small enough that these effects are negligible (probably corresponding to a radius considerably larger than the single cell size). Is that true? How reliable is the inferred force field?

- In Fig. 1d, the Golgi-to-nucleus angle is not considered, but the reason why does not seem to be explained in the main text. Why is that?
- I still do not completely understand what angles are used for the PCA. For example, in the cell orientation in Fig. 1b, the angles go from -90° to 90° . Does that mean that the authors do not only consider an orientation, but a proper director vector (corresponding to a polarity)? How is this director/polarity derived? For the PCA, do the authors use the director angle (between -90° to 90°)? Wouldn't it be more appropriate to use a quantity corresponding to $\cos^2(\text{angle})$ for the PCA? It is confusing that in Fig. 1b in the orientations, ellipses with very similar directions have very different colours.
- I have similar concerns as for the orientation of the Golgi-to-nucleus ratio. In particular, 1° and 359° correspond to almost the same direction (with a difference of 2°), yet wouldn't one consider a difference of 358° in the PCA. The same is also true in Fig. S3c where the angle now goes from -180° to 180° . But again, here -160° and 160° show in quite similar directions (with a difference of only 40°), yet their distance in the PCA will be 320° . Is that really the input in the PCA? In that case is it not very surprising that the angle does not contribute significantly to the PCA? Also, the range of angles should be consistent between Fig. 1 and Fig. S4 (and all other figures).
- Can the authors confirm that the PCA is done with standardised features? How is this standardization done for the jsPCA with multiple samples?
- The authors suggest that there are 3 clusters, but the AIC values for the case of 1 cluster and 3 clusters differ only by 0.6%. This seems like a tiny improvement, and it appears to be a classic example of over-fitting. If the authors prefer to keep the clusters in the manuscript, I would suggest that clusters be introduced for illustrative purposes to explain the range of behaviours in the morphospace.

Version 2:

Reviewer comments:

Reviewer #1

(Remarks to the Author)

The authors have now addressed my concerns.

Reviewer #2

(Remarks to the Author)

The authors have accounted for all of my comments.

The research is sound and the results support the conclusions. I believe this paper will significantly impact the field of heart development by introducing a data-driven approach to describe morphogenesis.

Dear Reviewers,

Before providing a point-by-point response to the reviewers, we would like to make the following general statement.

We first thank the two reviewers for their very useful comments. Their extensive feedback on our methodology led us to reevaluate our analysis pipeline. In particular, we found that, although being the natural tool for data exploration, classical PCA applied to cell morphological and mechanical features did not provide the most spatially informative representation of the cell features within the tissue samples. We quantified the Moran's Index (the spatial autocorrelation) of the Principal Components and found that PC1 had a low value, especially compared to PC2. To overcome the fact that classical PCA is not necessarily spatially informative, we turned to spatial PCA (sPCA, proposed in Jombart et al., 2008). This approach represents the data in a way that explains the most variance in the features (as in the classical PCA) but also represents the data in a spatially informative way, i.e. maximizing Moran's Index. In practice, sPCA is obtained by considering the product of the covariance matrix with Moran's Index. sPCA is specific for each tissue sample (because it uses the connectivity matrix of the network of cellular contacts when computing Moran's Index). Here, to be able to consider several samples, we developed joint sPCA (jsPCA) by jointly diagonalizing the sPCA matrices across the E9.5WT samples. The resulting eigenvectors with the highest eigenvalues are the axes explaining both the most variance within the cell features and the spatial coherence within the tissues. From this joint sPCA, we used the first axis as a basis for unsupervised clustering using a Gaussian Mixture Model. We chose this clustering approach because it enabled us to train the clustering on the E9.5 WT dataset and apply it to the other conditions.

We consider that this new methodological approach is more rigorous and captures the two main aspects of the data presented in this manuscript: the single-cell morphological and mechanical features on one hand, and the spatial distribution of these features within the tissue samples on the other hand. Although being a major refactoring of the analysis pipeline, this new methodological approach recovers the main biological contributions presented in the first version of the manuscript. More specifically, we identify from this work a new role for TBX5 in regulating the onset of patterned tension in the progenitor cardiac cell epithelium as progenitor cells segregate to alternate cardiac poles. Our results provide insights into the interplay between cell mechanical features and cell patterning during the morphogenesis of the heart tube, a particularly complex system.

Guijarro et al. Ms NCOMMS-23-34649

Replies to the Reviewers' comments

Reviewer#1:

Reviewer #1 (Remarks to the Author):

The manuscript by Guijarro et al explores the morphology of the apical side of the cells of the dorsal pericardial wall at early stages of heart tube formation in the mouse. They approach this problem by imaging and segmenting individual cells, which allows them to classify cells in an unsupervised manner, according to a selected set of geometrical parameters. Using this strategy, they classify groups of cells according to the most informative parameters for cell classification and report that cells close to the arterial pole present different morphologies and orientation than those close to the venous pole. Further to this, they study the expression of Tbx1 and Tbx5 in this context and characterize the changes in apical cell morphology in Tbx1 and Tbx5 mutants. While the approach undertaken in this manuscript is very attractive and potentially informative about developmental mechanisms, there are also important limitations that question the relevance of the observations. Overall, the conclusions on the mechanisms that govern heart tube morphogenesis are rather limited.

We thank Reviewer 1 for their careful reading of our manuscript, and considering our approach very attractive and potentially informative about developmental mechanisms. While we provide below a point-by-point answer to the reviewer's comments, we would like to first re-state our conclusions on the mechanisms that govern heart tube morphogenesis and the scope of our method.

First, our approach relies on morphometric and morphological features extracted from fluorescent microscopy imaging and a completely unsupervised analysis pipeline, therefore it is readily applicable on other epithelial tissues undergoing an interplay of mechanical and signalling dynamics.

Second, our approach is applied on the second heart field, where we are using unsupervised clustering on a spatially aware dimensionality reduction method to objectively divide the cells in the epithelium, and find the boundaries of the territories. This allows to find the signatures of the cells in the different clusters and find which of the geometrical and mechanical properties are involved in the partition. Moreover, by performing genetic perturbations, we identified a new role for Tbx5 in regulating tension in the progenitor cardiac cells, responsible to direct these cells to the venous.

Before the dorsal mesocardium breaks down the cells in this epithelium are under no stress, oriented and added to the arterial pole. Upon activation of Tbx5 in the SHF cells, there is ppMLC polarization that leads to orientation of the cells parallel to the antero-posterior axis and activation of anisotropic stress in the venous pole, corresponding to the cluster 3 in the clustering algorithm. This biological and morphological changes are guiding the cells to be incorporated to the venous pole of the heart.

When we disturb Tbx5, we lose the ppMLC polarization, and we lose the specific enrichment of cells corresponding to cluster 3 in the venous pole (cells parallel to the antero-posterior axis and no anisotropy). These results indicate that the expression of Tbx5 is necessary to establish the tension in the venous pole and consequently the

establishment of the Venous pole region necessary for a proper heart tube morphogenesis. In the revised manuscript we also show that loss of NKX2-5, associated with failure of heart tube extension at both poles, similarly leads to loss of posterior enrichment of cells from cluster 3 and relaxation of the progenitor cell epithelium. Together these observations suggest that it is the specific onset of SHF cell deployment to the venous pole that drives the establishment of patterned tension in the DPW during heart tube extension. We provide below point-by-point responses to the reviewer's comments.

Main points:

1. The study is based on acquiring a 2D projection of a 3D specimen, which distorts the real size and shape of the apical surface of cells. A 3D acquisition followed by a cartographic methodology to render realistic 2D apical surfaces would be required in this study. This is particularly important at the arterial and venous poles, where the dorsal pericardial wall cell sheet bends to adapt to the heart tube, but also at lateral sides of the specimens.

This point was also raised by Reviewer 2. In our analysis we have used DeProj to correct for tissue curvature and render realistic 2D apical surfaces (Herbert et al., 2021). We have emphasised this in the revised manuscript. DeProj is a tool made to correct for this distortion and allows us image the apical surface of cells in the dorsal pericardial wall in 3D. DeProj takes 1) the results of the segmentation on the projection - the green contour on the bottom-right quadrant above - 2) the height-map that follows the shape of the tissue - the gray smooth line on the top-right quadrant above - and "de-projects" the cell back on its original position in the tissue - in red, top-right quadrant. Thus, DeProj yields depth-corrected morphological measurements.

2. The study is based on the characteristics of the apical surface of the cells and does not consider the 3D geometry of the cells or the interactions of the cells with the

basement membrane. Based on several previous studies and the known behaviour of epithelia, the exclusion of this aspect limits the relevance of the study.

Our quantitative analysis focuses on the apical side of the cells as this epithelium is not a classical epithelium. It has previously been shown that while the apical side is epithelial-like in nature, the basal side has mesenchymal-like properties including dynamic filopodial projections (Francou et al., 2014). In particular, cell-cell junctions between cells in the dorsal pericardial wall, necessary for force transmission across the epithelium, are restricted to the apical side of SHF cells (Francou et al., 2014, Cortes et al., 2018). It is thus likely that mechanical forces acting on the epithelium result in shape deformation mainly on the apical side of the cells. In contrast, the basal, mesenchymal-like side is likely to be involved in exchange signalling with the extracellular matrix and basal lamina. We have added a brief description of the atypical epithelial features of the dorsal pericardial wall to the introduction (lines 98-101) of the revised manuscript and have noted that our analysis is restricted to the apical side in the discussion, as a potential limitation (lines 577-578).

3. The study claims the need of “Machine Learning” approaches to solve the problem of understanding “Morphogenesis”. Furthermore, they claim to have applied “Machine Learning” approaches and to have developed a software pipeline for the application of this “Machine Learning” approach. The reality is that the manuscript does not describe or apply any new “Machine Learning” approach but the use of well-established classification algorithms such as K-means. This is a very limited application of “Machine Learning” procedures.

Thank you for this comment. We have replaced the term “Machine learning” in the revised text, referring instead to the “unsupervised clustering method/algorithm.”, which is a more adapted description of our approach.

In this version of the manuscript, we have revisited our methodology and propose an original methodology termed joint spatial PCA. This builds on a previously published spatial PCA (Jombart et al. 2008) that is combined with a joint diagonalization of the various tissue sample score matrices. Unsupervised clustering is performed on this joint spatial PCA representation using the Gaussian mixture algorithm in order to train the unsupervised clustering on the data from E9.5WT embryos.

4. The parameters used for cell classification are few and rather redundant, such that in the end, the authors use nearly a unique parameter for cell classification.

This point was also by Reviewer 2 (point 3).

In this study we independently quantified nine parameters that are highly informative of the cellular heterogeneities observed in the tissue. One of the main objectives of this study was to be able to start with this set of features and identify the key parameters patterning the tissue. In that sense our study is a success as it indicates that the stress direction and cell orientation are the main parameters explaining the variation of cell features and their spatial distribution.

We would like to emphasize that stress direction and cell orientation, although sharing some aspects, are computed by two different methods (force inference and cell morphometrics respectively).

To address the relationship between the various parameters in a more systematic way, we have added Fig. S2 showing the correlation between the features. The correlation matrix shows that among the 9 features there is low correlation, except for cell area & perimeter that have a correlation coefficient (R) of 0.9 and cell orientation and cell stress direction with a correlation coefficient of 0.7.

5. The use of cell morphology to generate a map of forces is only an inference, but true forces or tensions are not measured. The limitation of this approach adds to the lack of characterization of the 3D structure of the cells and of their basal-side interactions.

Bayesian force inference has been validated in different epithelial contexts in the initial articles by Isihaira et al. 2012 and Kong et al. 2019. The robustness and precision of the method at the tissue scale goes beyond any other inference approach. Every cross-validation effort (in particular with ablation experiments) showed consistent results, in four different tissues, at very different scales, and in different animals. Hence, one can be confident that using the geometry of epithelial cells to infer stress patterns is reliable.

We have however consolidated the force inference results showing that there is higher tension in the posterior SHF with ppMLC localisation (Fig. 7). The dorsal pericardial wall is located behind the heart, and thus inaccessible for laser ablation-type experiment. Our previous Francou et al 2017 study used a low resolution circular wound assay to experimentally infer tension in the dorsal pericardial wall. Our force inference results support and significantly extend these results by providing a quantitative and epithelia-wide readout of tissue forces. We have highlighted these points in the revised discussion.

Concerning the basal-side interactions, please see our reply to Reviewer 1, point 2: it is the apical sides of these atypical epithelial cells that form the epithelial junctions and that change in response to epithelial tension.

6. The alternance of cell group identities around the arterial pole seems artefactual and derived from the bilateral symmetry or radial symmetry of the cell disposition around the outflow tract insertion site. The conclusions made about arterial pole rotation based on this classification are misleading. The authors measure angles with respect to a L-R

line and these scores cells oriented symmetrically with respect to the midline (or outflow tract insertion circumference) into different cell clusters, when, in fact, they are similarly oriented with respect to the embryonic symmetry references. The absolute values of angles with respect to the midline or to the closer tangent to the outflow insertion circumference should be used instead to determine the true classification.

This point has also been raised by Reviewer 2 in point 14.

To address this point, we have used the absolute values of cell orientation and cell stress direction. The results show that the optimal number of clusters is 2. They divide the cells into the anterior and posterior part.

This result has been added in Fig. S2f. From a biological point of view, it is interesting to keep the 3 clusters that show that the cells in the arterial pole alternate around the embryonic axis (or the outflow tract) in contrast to the situation close to the venous pole where no such alternation is observed. We have substituted the term rotation by alternation of clusters in the revised manuscript.

Other points

7. The study ignores proliferation of the cells under study. Related to this, what does group 4 from the K-means classification represent? Could this group represent dividing cells, considering their widespread distribution? This would be an interesting aspect of the study.

Use of the joint spatial PCA approach in the revised manuscript does not uncover a 4th group. However, proliferating cells have been shown to be enriched in the posterior region of the dorsal pericardial wall (Francou et al., 2018; Van den Berg et al., 2009).

8. The procedure to “clean” the distribution of cells belonging to the different clusters (Figure 2d) provides nicer images by neglecting the real data, but it is not justified by any biological argument.

We have now removed the filtering step when updating our methodological pipeline.

9. Line 335: “In Mef2cAHFCre; Tbx5fl/fl embryos cluster 3, distinguishing posterior DPW cells in wild-type embryos, is not observed (Fig. 5c)”. Actually, in Figure 5c’ a numerous group-3 cells can be observed.

In the revised manuscript we have added quantification of the number of cells belonging to clusters 1, 2 and 3 along the anterior to posterior axis. The result show that in E9.5 wild-type embryos cluster 3 cells are mainly located in the posterior DPW, close to the

venous pole, while clusters 1 and 2 cells are predominantly located anteriorly, close to the arterial pole. However, in Mef2cAHFCre; Tbx5^{fl/fl} embryos there is no significant difference in the distribution of cells in clusters 1, 2 and 3, that are uniformly distributed from the arterial to the venous pole (Figure 4d).

10. Line 363: "it can be observed that in presence of Tbx5, there is an enrichment of horizontal cells in the anterior DPW and an enrichment of cells oriented vertically in the posterior DPW (Fig. S9e)." Here, the authors are describing the wild type situation. Using "enrichment" to describe the wild type specimens is misleading.

We agree with the Reviewer 1 and have avoided the term "enrichment" in this context. We have quantified how many Tbx5+ cells have a cell orientation angle of $90^{\circ} \pm 10$ in the 4 conditions: wild-type E8.5, E9.5 and Tbx1 and Tbx5 mutant (Figure 5e).

11. Calls in the text to Figure 1 do not match Figure panels.

Thank you. Call-outs to Figure 1 (and all figures) have been corrected in the revised manuscript.

Reviewer #2 (Remarks to the Author):

This manuscript investigates the interplay between genetic factors and mechanical forces during heart development. It focuses on how progenitor cells from the second heart field (SHF) contribute to heart tube extension. Utilizing single-cell analysis, the researchers explore how cell shape and mechanical stress are related. The aim of this study is to generate a quantitative map of cell shape and mechanical features in of SHF progenitor cells and to study how different T-box genes, Tbx1 and Tbx5, affect these features.

The authors not only generate such a feature map of cells but find that Tbx1 guides cell orientation towards the arterial pole. On the other hand, Tbx5 activation aligns cell morphologies. These insights underscore the intricate connections between genetics and mechanical aspects in cardiac morphogenesis. The findings can have a significant impact on understanding not only heart development, but congenital heart defects.

Overall, the underlying idea of this study can have a significant impact on the field of heart morphogenesis and Evo-Devo in general. While for other biological systems, morphometric studies have already been done (e.g., Andrews et al. Development 2021

[27] or Viana et al., Nature 2023), to my knowledge there are no published studies that also consider force inference features, except a very relevant preprint by Hallou bioRxiv 2023. Hence, including mechanical features for a morphometric analysis is novel and a very promising idea for future studies of biological systems.

The idea behind the methodology of this study is in general sound but requires improvement (as pointed out below in the list of comments). My main concern is that it is not clear if advanced machine learning tools are really required for this data set that only uses 9 features and what value the PCAs/UMAPs provide.

Moreover, in its current form the manuscript lacks some explanation or at least a biological/biophysical hypothesis for the observed behaviour that goes beyond providing a map of cell features in the developing heart.

We thank Reviewer 2 for their positive assessment of our manuscript and in particular highlighting the interest of considering jointly single cell morphometrics and mechanical features. While we've considerably revisited our methodology, our major biological contributions remains the identification of a role for Tbx5 in establishing epithelial tension in the posterior second heart field close to the venous pole and inducing cell alignment and stress patterns in the progenitor cell field necessary for proper heart tube morphogenesis. In the revised manuscript we also show that loss of NKX2-5, associated with failure of heart tube extension at both poles, similarly leads to loss of posterior enrichment of cells from cluster 3 and relaxation of the progenitor cell epithelium. Together these observations suggest that it is the specific onset of SHF cell deployment to the venous pole that drives the establishment of patterned tension in the DPW during heart tube extension.

Below you will find comments that will hopefully help to improve this manuscript.

Major Comments:

1. Fig 1f-i and Fig. 2a-d proof that the cells are distributed in a continuum, disproving the existence of distinct clusters/populations (see also text line 198 for the PCA and text line 201-202 for the UMAP where this exactly is pointed out). What exactly are the clusters/populations that the authors find then? It might be better to treat it as a continuum and not as N distinct clusters.

The distribution of cell features forms a continuous population. Nevertheless, when performing a Gaussian mixture, we find that the Akaike information criterion is the

smallest when considering three clusters which indicates that this number of clusters provide an optimal representation. The clusters provide a coarse-grained representation of the distribution of cells. We are interested in their spatial localization within the tissue.

2. In Fig. S1a it seems like the tissue is curved. Is that taken into consideration when extracting the shape features and mechanics?

Yes, tissue curvature is considered in our analysis when we evaluate apical shape in 3D using DeProj depth-correction software. This is the same corrected data that we use for force inference. The deflection height of the tissue compared to its size is at most 13% which is an appropriate % to use Force inference. We have now highlighted this step in the revised manuscript. Please see our reply to Reviewer 1, Point 1.

3. Did the authors check that the 9 features are not correlated? Fig 1d seems to show that for example area & perimeter, cell orientation & stress direction and maybe cell neighbours & Golgi polarity are strongly correlated and usually one would do a PCA/UMAP with only one of them. Also, if one only has 6 to 9 parameters, it is not clear why dimensional reduction techniques are required at all. Two possible solutions I see are to extend the features studied to provide a more holistic description of the cell properties. Or reduce the number as features that are relevant as much as possible and do not perform the dimensional reduction.

Please see our response to Reviewer1, point 4.

Correlation analysis is shown in Fig. S3. Among the 9 features quantified there is low correlation, except for cell area & perimeter that have a correlation coefficient (R) of 0.9 and cell orientation and cell stress direction with a correlation coefficient of 0.7. In our revisited analysis pipeline, we propose first to compute the joint spatial PCA on the set of E9.5 WT tissue samples. We then use only the first joint spatial Principal Component to perform the unsupervised clustering. This principal component is composed of cell orientation and stress direction. Thus, as suggested by the Reviewer, we have thus reduced the number of features to the relevant ones for unsupervised clustering.

4. It is unclear to me why the authors perform the UMAP and dimensional reduction in general. The main findings about the different cell populations/clusters are mostly disconnected from the data. For example, what insights does Figure 5f provide? Please also note this recent paper (Chari, Pachter, Plos Comp Biology 2023) which shows that UMAP is not a good method for cluster verification.

We removed the UMAP plots to avoid confusion. We perform the joint spatial PCA on the initial set of cell features (before any dimensionality reduction). We used the Akaike

Information Criterion to identify the relevant number of clusters for Gaussian mixture on the first jsPCA.

5. Clustering using the k-means algorithm can be improved. My concerns are that k-means in some special circumstances can bias towards equally sized clusters. A. Can the authors verify their results using other clustering algorithms? B. Also, when performing the k-means clustering, how do the authors define the initial points of the k-means clusters. How many initial conditions did the authors try out and how robust is the calculation of the ideal cluster number?

We changed our unsupervised clustering to Gaussian mixture, such that we are able to train the categories on data from the E9.5 wildtype embryos and use the trained clusters on the other conditions. The optimal number of clusters was obtained by minimizing the Akaike Information (AIC). As shown in Fig.S4.

To check the reproducibility of the clustering we performed 100 independent iterations of Gaussian mixture clustering. When comparing these 100 iterations, we found that, among all the pairs of cells, only 2.65% are not always or never clustered together.

6. Finding the individual clusters 1, 2 and 3 is one of the main highlights of the authors manuscript. My main concern is that these clusters are apparently defined several times and for different underlying data set (e.g., for Fig. 2 the authors use E9.5 embryos, for Fig. 24 the authors use different time points, for Fig. 5 the authors use mutants). But how can the authors compare the different clusters if each time the authors perform a different k-means clustering? Shouldn't the authors only define the clusters once for the WT and then classify the cells in the other conditions using the WT training data? For example, Fig. 2e-h and Fig. 5e show qualitatively very similar distributions. Additionally, it would be helpful if the authors could give the clusters specific names and call them "xyz population". It is confusing to call them clusters that in Fig. 3d''' form also distinct spatial clusters. Better to say that cells of population xyz (the orange ones) form two distinct clusters in Fig. 1d'''. Also, often it is not clear which data was used to define the so-called cluster 3, e.g., in line 354.

Following Reviewer 2 suggestion, we have defined our clusters on E9.5WT condition, and used these learned categories to classify cells in the other conditions. We prefer to maintain the names Cluster 1, 2, 3 to avoid any bias in the analysis.

7. If the authors need to consider different conditions for the clustering or dimensional reduction of cells, how do they weight the different cases? E.g., if the authors compare

1000 cells of the WT and only 100 cells of mutant tissue, shouldn't one weight the data accordingly so that each condition contributes equally??

Thank you for this relevant comment. Now that the clustering is learnt on the E9.5 WT data and applied to the other conditions, we don't need to weigh the different cases.

8. Is it possible and feasible to make proper mechanics measurements (e.g., laser ablation experiments) to check if the force inference results/stress maps are correct?

Please see our response to Reviewer 1, Point 5. We have validated force inference results with ppMLC polarization in Fig. 6.

Other

Comments:

9. Line 41: It might be good to say what Tbx1 is in the abstract.

We have introduced Tbx1 and Tbx5 in the abstract as suggested. Line 43-44

10. Line 91 and 430: The concept of CRNs is not really established and needs more explanation in the manuscript. In general, CRNs as a concept seem to be badly named since genes and gene expression are part of cell biology and one would assume that GRNs are just a sub-group of CRNs.

We have clarified our use of the term Cell Regulatory Networks in the revised manuscript. Line 479.

11. Line 130: of course, image analysis is non-invasive. I guess the authors mean non-invasive force measurements based on image analysis. Also, would the authors call the imaging/microscopy they perform non-invasive?

Thank you for this point. We have substituted the use of the term non-invasive by the term image analysis inferred forces method in the revised manuscript.

12. Line 169: the number of cell neighbours is not a morphological feature!

We don't present the number of cell neighbours as a morphological feature, but as a characteristic of cell shape within an epithelium (Gibson et al. 2006).

13. Fig. 2e-h and Fig. 5e: Unclear how the distributions are normalised. Clearly, the area is not the same for all curves. Best is to show the probability density function.

We are now using the probability density function kdeplot with seaborn library in python. We have substituted the plots by histogram plots from seaborn library and used the normalize statistical function.

14. Line 170: The angle between the centre of the nuclei to the centre of the Golgi, relative to the embryonic left-right axis goes from 0° to 360°. Are these also the numerical features the authors use for the analysis? If yes, that would mean that the numerical features are very different for 1° and 359°, while the directions are very similar?

Golgi is not included in the single cell morphometric analysis. PCA analysis of the features showed that Golgi accounts only for a very small proportion of the data variance (the 9th PCA is associated mainly to Golgi orientation and accounts for 1% of the total variance). For orientation the angles go from 0-180°, the angle histograms are in general "far" from the boundaries of angle definition.

15. Line 200/201: Fig S2 is supposed to show reproducible behaviour among all embryos, but indeed does not show anything in this direction. But in fact, it would be interesting to show the morphospace and feature distributions for all individual embryos.

Thank you for this suggestion. In this version of the manuscript we have not represented the embryos in the joint spatial PCA nor classic PCA. Supplementary figures grouping all the embryos used for the single-cell morphometrics analysis are available for each genetic and temporal condition (Fig. S8-S13, S15). We see reproducible behavior for the spatial distribution of cluster 3 in each condition.

16. Line 223: Why exactly do the authors use a UMAP and not PCA?

We are now using PCA and spatial PCA.

17. Line 225: I understand that the authors want to reduce imaging noise. But why would the authors want to reduce biological noise, while this could also provide useful insights? It might be interesting to look at biological noise.

We have now removed the filtering step when updating our methodological pipeline. The variation of cell features is shown the PCA analysis and the diversity of results is shown across the various tissue samples see supplementary Figs. S9 to S13, S15.

18. Line 231-236: Is the ideal cluster number 3 for all individual embryos?

Using the Akaike Information Criterion (AIC), we find that the ideal cluster number is 3. This is shown in revised Fig. S4

19. Line 241-244: How do the authors find these angles? Are these the exact numbers extracted from the data?

We performed the mean and the standard deviation of the values for the cells belonging to each of the clusters. The standard deviation values have been added to the revised text for clarity.

20. Fig. 4: Can you show each channel individually? Right now, the composite image is very hard to see any co-expression. Overall, I have problems linking text lines 295-305 to Fig. 4 and it would be good if the figure can be improved so that it is easier to understand in which regions exactly to look.

We have addressed this point by adding a summary schema to the revised Fig. 3. In addition we have added a quantification of the relative pixel intensity of Tbx1 and Tbx5 probes, showing transient coexpression and have added views showing each channel separately in a supplementary figure for clarity (Fig. S6).

21. Is it possible for the same cells to look at shape, mechanics and Tbx1 and Tbx5 expression? If that is experimentally feasible, it would be interesting to quantify individual gene expression of cells.

Coupling spatial gene expression patterns with our single cell morphometric analysis is a challenge we aim to address in future experiments. Using immunofluorescence we cannot score both TBX5, TBX1 and the different parameters we are evaluating due to limited channel numbers. Technically, it is not possible to visualize transcripts of the 2 genes plus a membrane marker for cell segmentation as In situ hybridisation techniques disaggregate the membrane proteins during the proteinase step. What we have done is used the Tbx5 antibody and membrane markers to identify the Tbx5+ cells and, based on our observations using RNAscope, label cells negative for Tbx5 as Tbx1+ by default.

22. Line 329: I am not sure how I am supposed to see this from Fig. 5f alone.

Both Reviewers have raised the point about the interest of the UMAP projections. The UMAP projections have been removed from the revised manuscript as they do not contribute to the revised data analysis pipeline.

23. Line 334-335: The data does not seem to show that the percentages are the same

as in the WT. Could you show that the observed difference in percentages is not significant?

In the revised manuscript, with the new pipeline the percentages of the clusters are the same among the different genetical and temporal conditions. What changes is the distribution of the cells labelled as cluster 3 as shown in Fig. 5.

24. Fig 6 and from line 351: Could the authors show in the figure where the specific clusters are to make it easier to follow the text?

We have indicated the location of specific clusters in the revised figure to facilitate reading.

25. Line 382 – Line 389: Could this be explained by a too low concentration of BMS493?

This is one possibility, potentially linked with the fact that embryos are developing very rapidly at this timepoint and not all embryos in a litter have the same exactly stage on exposure to BMS493.

26. Line 430 – 431: Turing did not consider mechanics in his theory of mechanics, but interactions of different biochemical cues with different diffusion coefficients. On the other side D'Arcy considered the role of mechanics on morphogenesis, but with no focus on biochemistry.

We have modified this section of the Discussion and added reference to D'arcy Thomson in the revised text, as suggested.

27. Line 441 and 485: I am not sure that it is appropriate that the authors addressed the regulatory networks. They did not analyse the feedback or proper interactions between the individual features.

Thank you, we have clarified this point in the revised text.

28. Figure 5a'-c': Why do the authors not show their connectivity graphs here with the "cleaned" cluster distributions?

We have added cluster distributions to revised Fig. 6, as suggested by the Reviewer. However, the filtering step has been removed from our data analysis pipeline.

29. Fig. 6a'' and similar figures: The angles are continuously distributed. Hence, the

probability to find exactly an angle of 90° is zero and the authors need to define a range they consider defining the red cells. What is this condition?

We have clarified the range of angles used ($90^\circ \pm 10^\circ$) here and elsewhere in the revised manuscript.

Typos and small errors:

30. Fig. 1b and other figures: Color-coding cell orientation and cell eccentricity like the cell area might look better. Additionally, some figures have a very low resolution (e.g. Golgi figure in Fig. 1b). But this might have happened during the submission and does not need to be the fault of the authors.

Cell orientation has been color-coded following a cyclic gradient colour palette due to the cyclic nature of the cell orientation data. This colour palette is more suitable than the continuous palette that has been used for area and perimeter as explained in the misuse of colour in science communication (Crameri et al. 2020).

31. Fig. 1b and 1c and all other figures: colorbars need units. Also, in Fig. 1c, the cell pressure has no colorbar. Another example is Fig. 1e where it is not shown what the the y-axis is.

Thank you, we have corrected these points in the revised manuscript. Pressure in this case has no units as you can see in the force inference paper of Ishihaira et al, 2012 and Kong et al 2019. The value is a normalization with the pressure of the tissue.

32. Line 121-122: provide a reference.

Two references have been added to the revised text: Francou et al and Li et al.

33. Line 192: Do the authors mean Fig 1D?

Yes, Fig.1 d, we have updated the reference in the text.

34. Line 198-199: What is X and Y?

In the corrected version of the manuscript we have removed this figure.

35. Line 199: I can't find Fig. 1k.

This has been corrected to Figure 1 f-i in the revised manuscript.

36. Line 201: Unclear what is meant by "Linear distribution"?

Linear distribution has been changed to continuous distribution.

37. Line 203-208: This text is written very unclear. What does it mean that cell elongation is different on the left and the right side? And what are gradients according to apical area, perimeter, and pressure? Also, Fig 1h'-j does not show what is written in the text and 1j does not exist.

Thank you, we have modified this point and substituted by: Since the aim of jsPCA is to obtain orthogonal directions of variations, we expect the patterns associated to jsPC1 (cell orientation and stress direction) and jsPC2 (cell area and perimeter) to be quite different, as represented on Fig. 1f,g. (line 263-264)

38. Sometimes figures do not appear in the right order, e.g., Fig. 3d''' (line 268) is shown before 3d''. I suggest the authors go again over their figures and the text and see how they can improve the synchronicity between them.

The logical flow of figure panel call-outs has been verified in the revised text.

39. Line 270-271: This sentence is not clear to me. Where is this shown?

In Fig.3a, Fig. 3c-c'''. This has been updated in the text.

40. Line 272: What means above?

The word "above" has been deleted from the revised text.

41. Line 433: What is the final architecture of an embryo? An embryo is a transitional configuration of cells and to a (probably very limited degree) the adult is the final architecture, not?

We agree and have deleted "embryo" from the revised text.

42. Line 465: typo "including"

Corrected.

43. Line 476: typo “deponedence”

Corrected.

44. Line 476-Line 479: This sentence is very hard to comprehend.

We have clarified this sentence in the revised manuscript.

45. Line 541-542: How is the connectivity graph a mathematical representation of a CRN? It is unclear to me what the CRN is in that case? Is it a network of the different features or different cells?

According to Martinez Arias (Martinez Arias et al., 2021): “The core of CRNs are protein-protein interactions. Thus, the nodes of CRNs would be the components of cytoskeletal, adhesion and traffic related molecules”.

The phalloidin stains the actin filaments (F-actin). In the case of the SHF cells, F-actin accumulates at the border of the cells, as well as other proteins that regulate cell mechanics such as ppMLC. In the case of the SHF cells, the proteins (F-actin and ppMLC) assemble to maintain the integrity of the structure and create the graph of tensions of the epithelium. And the connectivity graph is the graph that connects only the cells that share membrane protein-protein interaction. The node is the cell and the edges are the protein-protein connection.

46. Line 594-595: Isn't a circle the same as an ellipse with identical short and long axis? Then how can 0 be a rounded shape (circle?) and 1 be an ellipse?

The eccentricity has been calculated in the Deproj software (Herbert et al., 2021). The computation is based on the classical eccentricity definition $e = \sqrt{1 - \left(\frac{b}{a}\right)^2}$, where a is the long axis and b the short one. If we consider a circle, we have $a = b$ and $e = 0$. All values of e such that $0 < e < 1$ correspond to ellipses with different long over short axes ratios. And $e = 1$ is the extreme case of infinite elongation, where it is no longer an ellipse but a parabola.

47. Line 608: Before the authors said it was xlsx files, what are the csv files?

We have now converted all the embryos in a single .csv file.

48. Fig. S2: typo "Dimentionality"

Corrected

49. Fig. S3a: Why are there two cell areas?

Corrected.

50. Fig. S6e: Figure seems to be cut.

Corrected.

Reviewer #1 (Remarks to the Author):

1.- In my opinion, the results of the deformation/tension maps indicate that there are two colliding stresses; one tangential to the outflow track insertion circumference and another in the A-P direction. The A-P influence is predominant posteriorly, but it is not exclusive of this region, given that group-3 cells can be seen as well in anterior regions. And vice versa, groups 1 and 2 also show posteriorly located cells intermingled with group-3 cells. The outflow track influence is obviously more predominant anteriorly. In my opinion, the way clusters are defined and explained in the manuscript fails to explain this organization. In particular, the maintenance of two groups for the anterior region is particularly problematic as this is derived from measuring cell deformation in the A-P direction, whereas the organization of the tissue in this region follows a radial symmetry. The maintenance of these two anterior groups may mislead the reader into thinking that there is a L-R difference. In my opinion, the subdivision in two clusters shown in Figure S4b is cleaner and much more descriptive of the biology of the system. Readers would benefit from seeing the changes that take place in these two clusters in *Tbx1* and *Tbx5* mutants.

We thank the Reviewer for this valuable point. In the revised manuscript we have now included subdivision of the cells into two clusters computed with orientation and stress orientation symmetrized with respect to the AP axis (as in original Fig. S4b). The distribution of cells into two clusters has been added to Fig. 2k and the morphometric map for all embryos in Figs. S5, S6, S9 and S10. We have also maintained the three clusters visualization as this approach is based on the entire set of raw features derived from the morphometric measurements, and as such, is as unsupervised as possible; this approach can thus be directly applied to various 2D systems beyond AP symmetric ones. However, as the Reviewer points out, including the two cluster analysis based on symmetrized angles and stress direction helps interpretation of the system and the dynamic of the stresses. In particular, with reference to Fig. 2k we have modified the text to highlight that symmetry around the outflow tract is radial, as noted by the Reviewer. We have also modified the discussion on page 17 to point out that this appears to be the result of epithelial stress that is independent from the TBX5-dependent tension operating in the posterior region of the epithelium, consistent with the changes observed in *Tbx1* and conditional *Tbx5* mutant embryos. Please also see our reply to Reviewer 2, point 6.

The text has been modified as follows:

Page 9: "Given the bilateral symmetry of clusters 1 and 2 we also classified cells into two clusters computed with cell orientation and stress direction symmetrized with respect to the anterior-posterior axis (Figs. 2k and S4). Although introducing a priori knowledge about the symmetry of the system, and hence some level of supervision, this two cluster analysis highlights the radial symmetry of cells in the anterior DPW as they converge on the arterial pole."

Page 10: "Classification of E8.5 cells into two clusters, computed with symmetrized cell orientation and stress direction, also revealed that distinct anterior and posterior clusters emerge between E8.5 and E9.5 (Fig. S6)."

Page 12: " Classification of *Tbx1*^{-/-} and *Mef2cAHFCre;Tbx5^{fl/fl}* cells into two clusters, computed with symmetrized cell orientation and stress direction, revealed that while two regionalized clusters are maintained in *Tbx1*^{-/-} embryos, emergence of a distinct posterior DPW cluster fails in *Mef2cAHFCre;Tbx5^{fl/fl}* embryos (Figs. S9, S10). In addition, loss of a distinct anterior cluster

after symmetrization in *Mef2cAHFCre;Tbx5^{fl/fl}* but not *Tbx1^{-/-}* embryos suggests that radial symmetry resulting from convergence of cells on the arterial pole does not depend on TBX5."

Page 17: "Consistent with this conclusion, symmetrizing cell orientation and stress direction allowed cells be classified into two clusters distributed anteriorly and posteriorly at E9.5 but not E8.5. Moreover, loss of regionalized distribution of these clusters in *Mef2cAHFCre;Tbx5^{fl/fl}* embryos points to at least two independent forces driving epithelial stress in the DPW: TBX5-dependent patterned tension close to the venous pole and TBX5-independent stress as cells converge bilaterally on the arterial pole."

2.- The establishment of a particular threshold to determine that anisotropy does not exist in a cell is arbitrary and the chosen value (0.3) appears too low to consider that measurements above this threshold are isotropic. Apparently, this threshold is set at 0.08 for figure 6 (See line 656 and Figure legend) but I think this is a mistake, given that it is set at 0.3 in Figure S5 and all the rest of Figures with the same type of representation. In any case, it would be better to show a continuous color scale reflecting all measurements and their distribution in the tissue for all specimens (at E8.5 and E9.5). This would also show the anisotropy map independently of the stress direction.

Thank you for this point. The use of a threshold to illustrate regional anisotropy has been removed and instead the figures have been substituted by a continuous color scale of the measurement of the anisotropy ranging from 0 to 1. This format has been used for all specimens in the main and supplementary figures.

3.- It would be important for readers to see the distribution of each cell cluster in every specimen, as shown in Figure 2b for a single specimen.

Four additional supplementary figures (Figs. S16, S17, S18 and S19) have been added showing the distribution of each cluster in all embryos analyzed.

4.- Authors should explain better what was measured and statistically analysed in Figure 6e and 6f.

Figs. 5d, 6e and 6f show the distribution of cluster 3 cells along the antero-posterior axis of the dorsal pericardial wall (represented on the Y-axis), 1 being the arterial pole, -1 being the venous pole and 0 the mid-point of the DPW. The statistical difference of the mean values was assessed by Mann Whitney U test as explained in the Materials and Methods - Statistics and reproducibility section. For clarity, the following has been added to the legend of Fig. 5d:

"d. Boxplot quantification of the spatial coordinates of cells belonging to each of the clusters along the antero-posterior axis (with y coordinates normalized from -1 (VP) to 1 (AP)) for E8.5 and E9.5 wild-type (WT), *Tbx1^{-/-}* and *Mef2cAHFCre;Tbx5^{fl/fl}* embryos; p-values are based on Mann Whitney U test and test whether the spatial distribution of a cluster is closer to the Venous Pole. Scale bars: 50µm."

The legend of Fig. 6e-f has also been modified:

"e. Boxplot quantification of the spatial coordinate of cells oriented vertically (90°±10) or horizontally (0/180°) along the anterior-posterior axis (with y coordinates normalized from -1

(VP) to 1 (AP)); p-values are calculated using a Wilcoxon signed-rank test. f. Boxplot quantification of cells with anisotropic stress (< 0.3) along the antero-posterior axis (with y coordinates normalized from -1 (VP) to 1 (AP)). p-values are calculated using a Wilcoxon signed-rank test and test whether the spatial distribution of cells deviates from a symmetric distribution. Error bars represent standard deviation. Scale bars: $50\mu\text{m}$."

5.- In Figure S1, the authors describe their 3D analysis of the Golgi-to-nucleus orientation measurements, however, they do not specify the thresholds used to determine the polarity. Unfortunately, the color scale cannot be properly visualized in the graph shown in Figure S1D (below) and this should be improved.

Golgi position was calculated relative to the center of the nuclei of the cell. The center of nucleus C (0,0,0) is the threshold that have been used to established the anterior-posterior (y axis), left-right (x axis) or apical-basal (z axis) position of the center of the Golgi. If the values were superior to this threshold in the respective axis, they were considered anterior, right or apical. On the contrary if the values are below this threshold, the Golgi was considered posterior, left or basal with respect to the x,y,z axes. Figs. S1c and d have been modified to clarify this point and better visualize the results.

Reviewer #2 (Remarks to the Author):

This manuscript presents a comprehensive analysis of the role of T-box genes in regulating epithelial tension during cardiac morphogenesis. The study utilizes advanced single-cell morphometrics and force inference techniques to elucidate the dynamic interplay between genetic and mechanical factors in heart tube extension. The findings offer significant insights into the molecular mechanisms underlying heart development and congenital heart defects.

The authors have made significant improvements to the previous version of the draft. Especially using the spatial PCA seems very appropriate and provides useful insights into studying SHF.

I believe this manuscript can provide significant contributions to the study of heart development and their methodology will also be very useful for studying other 2D tissues. Yet, I have some concerns that need to be accounted for before I can give my final recommendation for this paper:

1• The author's reply to Reviewer 2 (major comment 2) and the methods section imply that the force inference was done on depth-corrected data. However, Figure 1a does not really illustrate that the force inference was applied to the depth-corrected data. Would the depth correction not lead to distortions that will also affect the inferred force field? I expect that the curvature is small enough that these effects are negligible (probably corresponding to a radius considerably larger than the single cell size). Is that true? How reliable is the inferred force field?

We thank the Reviewer for making this important point - we realize that there was an error in our prior response and that force inference was indeed not calculated on depth-corrected data, as the Reviewer surmises. This in fact avoids distortions from depth correction that might affect the inferred forces. To address the comments of the Reviewer we have calculated the radius of curvature of the tissue for both the first and second principal curvatures (r_1 and r_2). We show in a histogram plot that the radius of curvature is significantly bigger than the radius of the cells indicating that the curvature of the tissue is small enough to have only a negligible effect on

force inference calculations. We have added this analysis to the revised manuscript (Figure S20).

2• In Fig. 1d, the Golgi-to-nucleus angle is not considered, but the reason why does not seem to be explained in the main text. Why is that?

We have quantified and reported Golgi-to-nucleus angle in Fig. S1. The conclusion of our analysis is that the Golgi-to-nucleus angle's most relevant direction of variation is in the Apico-basal direction, and is therefore not relevant for the spatial organization of the tissue with respect to the other apical features measured. We have clarified this in the revised text and to avoid confusion, we have removed Golgi-to-nucleus angle from the main set of features in Fig. 1.

3• I still do not completely understand what angles are used for the PCA. For example, in the cell orientation in Fig. 1b, the angles go from -90° to 90° . Does that mean that the authors do not only consider an orientation, but a proper director vector (corresponding to a polarity)? How is this director/polarity derived? For the PCA, do the authors use the director angle (between -90° to 90°)? Wouldn't it be more appropriate to use a quantity corresponding to $\cos^2(\text{angle})$ for the PCA? It is confusing that in Fig. 1b in the orientations, ellipses with very similar directions have very different colors.

The cell orientation measurement ranges from 0° to 180° . For clarity this is now clearly indicated on the orientation maps for every figure using a color code rather than ellipses. For the PCA, the cell orientation and stress direction are, as all the other features, standardized. In addition, we also now include the results of clustering where the orientation and stress direction have been symmetrized with respect to the anterior posterior axis, i.e. we took the absolute value of the orientation - 90° , and the absolute value of the stress orientation instead of the original angle value (Figs. 2k and Figs. S5, S6, S9 and S10). Please see our reply to Reviewer 1 point 1.

4• I have similar concerns as for the orientation of the Golgi-to-nucleus ratio. In particular, 1° and 359° correspond to almost the same direction (with a difference of 2°), yet wouldn't one consider a difference of 358° in the PCA. The same is also true in Fig. S3c where the angle now goes from -180° to 180° . But again, here -160° and 160° show in quite similar directions (with a difference of only 40°), yet their distance in the PCA will be 320° . Is that really the input in the PCA? In that case is it not very surprising that the angle does not contribute significantly to the PCA? Also, the range of angles should be consistent between Fig. 1 and Fig. S4 (and all other figures).

Golgi-to nucleus polarity has been removed from Fig. 1 as it has not been used in the PCA analysis. Please also see our reply to Reviewer 1 point 5. Figs. S1c and d have been modified to clarify this point and better visualize the results.

5• Can the authors confirm that the PCA is done with standardised features? How is this standardization done for the jsPCA with multiple samples?

The PCA is performed with standardised features. In the case of jsPCA with multiple samples, the standardization is done sample by sample. This is consistent with the procedure of joint diagonalization. Each sPCA matrix is normalized separately but diagonalized jointly.

To clarify we have updated the text in the Methods, section “Data analysis for single-cell morphometrics” with the following sentence: “Each column of the features matrices are centered and scaled for each embryo separately.”

6• The authors suggest that there are 3 clusters, but the AIC values for the case of 1 cluster and 3 clusters differ only by 0.6%. This seems like a tiny improvement, and it appears to be a classic example of over-fitting. If the authors prefer to keep the clusters in the manuscript, I would suggest that clusters be introduced for illustrative purposes to explain the range of behaviours in the morphospace.

We have used AIC to confirm that choosing 3 clusters was consistent with the structure of the data. As noticed by the reviewer, the variation between 1 cluster and 3 clusters is very small. We agree that the cluster analysis is very helpful to explain cell orientations in morphospace and thus the stresses acting on the epithelium. While having only 1 cluster would be uninformative, with 2 clusters (unsymmetrized) bilateral asymmetry dominates (see Fig. R1). We therefore concluded that 3 clusters would be the most informative value. Similarly, we chose two clusters as being the most informative in the case of AP-symmetrized stress direction and orientation.

Fig. R1. E9.5 wild-type cells after classification of unsymmetrized data into 2 clusters plotted back in their physical space and color coded with cluster 1 cells in green and cluster 2 cells in purple. This highlights left right symmetry and the posterior enriched cluster of cells observed after classification into 3 clusters is not observed.